# Medicinal plants for allergic rhinitis: A systematic review and meta-analysis

Xin Yi Lim[1]*, Mei Siu Lau[1], Nor Azlina Zolkifli[1], Umi Rubiah Sastu@Zakaria[1], Nur Salsabeela Mohd Rahim[1], Nai Ming Lai[2,3], Terence Yew Chin Tan[1]

1 Herbal Medicine Research Centre, Institute for Medical Research, National Institutes of Health, Ministry of Health, Setia Alam, Malaysia, 2 School of Medicine, Taylor's University, Subang Jaya, Malaysia, 3 School of Pharmacy, Monash University Malaysia, Bandar Sunway, Malaysia

* limxinyi.lim@gmail.com

## Abstract

Herbal medicine is popularly used among patients who suffer from allergic rhinitis. This systematic review and meta-analysis was conducted to evaluate the efficacy and safety of single medicinal plants in the management of allergic rhinitis. We searched MEDLINE, CENTRAL, and Web of Science for randomised controlled trials which evaluated the use of single medicinal plant for allergic rhinitis among adults and children. Twenty-nine randomised controlled trials (n = 1879) were eligible while 27 (n = 1769) contributed data for meta-analyses. Most studies (studies = 20) compared medicinal plants against placebo and *Petasites hybridus* was most frequently investigated (studies = 5). Very-low-to-low-certainty evidence suggests that compared to placebo, single medicinal plants may improve overall total nasal symptoms (SMD -0.31, 95% CI -0.59 to -0.02; participants = 249; studies = 5; $I^2$ = 21%) especially nasal congestion and sneezing; and rhinoconjunctivitis quality of life (RQLQ) scores (MD -0.46, 95% CI -0.84 to -0.07; participants = 148; studies = 3; $I^2$ = 0%). Moderate-certainty evidence show no clear differences between single medicinal plants and antihistamine in overall symptoms (Total nasal symptoms: SMD -0.14, 95% CI -0.46 to 0.18; participants = 149; studies = 2; $I^2$ = 0%). As adjunctive therapy, moderate-certainty evidence shows that medicinal plants improved SNOT-22 scores when given as intranasal treatment (MD -7.47, 95% CI -10.75 to -4.18; participants = 124; studies = 2; $I^2$ = 21%). Risk of bias domains were low or not clearly reported in most studies while heterogeneity was substantial in most pooled outcomes. Route of administration and age were identified to be plausible source of heterogeneity for certain outcomes. Medicinal plants appear to be well tolerated up to 8 weeks of use. Clear beneficial evidence of medicinal plants for allergic rhinitis is still lacking. There is a need for improved reporting of herbal trials to allow for critical assessment of the effects of each individual medicinal plant preparation in well-designed future clinical studies.

**Data Availability Statement:** All relevant data are within the paper and its Supporting Information files.

**Funding:** The author(s) received no specific funding for this work.

**Competing interests:** The authors have declared that no competing interests exist.

## Introduction

Allergic rhinitis is a highly prevalent health condition affecting people of all ages worldwide. In Asia, the prevalence of allergic rhinitis was reported to range from 1.15% to more than 50% between 1994 to 2017. Higher prevalence has been seen especially among populations who were frequently exposed to triggering allergens due to geographical or work-related factors [1]. Patients suffering from allergic rhinitis often present with one or more classical symptoms of nasal obstruction, rhinorrhoea, sneezing, and nasal itching; sometimes accompanied by eye symptoms such as red, watery, or itchy eyes [2]. Thought to be Immunoglobulin E (IgE)-mediated, allergic rhinitis is primarily an allergic-origin chronic inflammatory condition [3]. Despite being a common health issue faced by many, the impact of allergic rhinitis is often overlooked, as its symptoms are not particularly debilitating in majority of the cases. Though not experiencing serious symptoms, many people who suffer from allergic rhinitis require long term medications to control the disease and yet, still have a significantly reduced quality of life in aspects of work and daily functions [4].

The Allergic Rhinitis and its Impact on Asthma (ARIA) guideline published in 2019 recommends a multidisciplinary, evidence-based approach in management of allergic rhinitis, based on the Grading of Recommendations, Assessment, Development, and Evaluation (GRADE) evidence assessment and includes integrated care pathways. Overall, a step-up step-down approach is adopted consisting of mainly oral antihistamines, intranasal corticosteroids, and/ or a leukotriene receptor antagonist, first as a single agent and then combined when needed depending on the severity and control of symptoms. ARIA also recommends to take into account patients' empowerment and preferences when devising a care plan [5]. The rationale for such an approach is supported by findings from real world evidence which show that patients often do not follow the treatment plan provided, prefer to self-treat, and have a low adherence to prescribed treatment [6]. In fact, there is compelling evidence that many who suffers from allergic rhinitis are interested in seeking alternative treatment. A prospective study in Turkey found that a significant proportion (37.3%) of patients with allergic rhinitis had used natural products or herbal-based therapies at least once in attempt to control their disease [7]. In Malaysia, 62.1% of allergic rhinitis patients from two public hospitals reported using complementary and alternative therapy within the past ten years [8]. The popularity of such therapies in allergic rhinitis is further acknowledged by the inclusion of dedicated sections addressing roles of complementary and alternative medicine in several well-established clinical practice guidelines [9, 10].

Several systematic reviews on complementary and alternative medicine for allergic rhinitis have been conducted, mostly focused on Traditional Chinese Medicine which consist of mixtures of oral and topically applied Chinese herbs, moxibustion, and/or acupuncture [11–14]. A recent systematic review by Hoang et al. on herbal medicine for allergic rhinitis which collated evidence of herbal medicine use (as single herbs, mixture of herbs, or in combination with procedure-based therapies, including Traditional Chinese Medicine) from 32 randomised controlled trials was published in 2021 [15]. In the research and implementation of herbal medicine, the definitive role of single medicinal plants can often become obscured due to the concurrent use of mixtures and other therapies. In addition, the use of Chinese herbal medicine is based on a set of distinctively different physiological and treatment principles when compared to contemporary conventional medicine, which is the most widely used approach globally. Specifically for herbal interventions, critical appraisals on the details and quality of investigational herbal products provide pivotal information when evaluating evidence derived from herbal trials. In published allergic rhinitis clinical practice guidelines, limitations in general study design have been cited as some of the main contributors to ambiguity in evidence

strength when making recommendations [9, 10]. This pivotal information was not addressed by the latest review conducted by Hoang et al. [15], though it did show the rising trend of herbal medicine clinical trials for allergic rhinitis when compared to an earlier systematic review conducted in 2007 [16]. Considering current literature gap, this systematic review was carried out to evaluate specifically the efficacy and safety of single medicinal plants in the management of allergic rhinitis from randomised controlled trials.

## Methods

### Study eligibility

The review team followed pre-set inclusion criteria to identify and include randomised controlled trials relevant under this review's objective with the following Study design and Population, Intervention, Comparison, Outcomes (PICO) elements. Additional exclusion criteria are applied to clarify the inclusion limits for 'Intervention', 'Outcomes', and 'Study design' and are presented below.

**Study design.** Only randomised controlled trials are included as we aimed to meta-analyse the highest level of available evidence. Articles that presented evidence on isolated phytochemical compounds were excluded as such evidence do not adequately represent whole medicinal plant extracts.

**Population.** Patients (adults and children) with clinically diagnosed allergic rhinitis of:

- All subtypes and severity

- Not undergone corrective surgery

- Without active infection

- With or without co-morbidities

The diagnosis of allergic rhinitis should be clearly stated in included papers. Diagnosis should be made based on characteristic clinical symptoms of allergic rhinitis, detailed clinical history and examination, with or without additional allergen testing.

**Intervention.** Herbal medicine as a single medicinal plant:

- As the sole intervention or as add-on therapy to conventional/regular treatment

- In all formulations (e.g., fresh, dried, extract, decoction, etc.)

- In the form of oral or intranasal administration (non-external, systemic administration)

- For a duration of at least one week or longer

**Comparator.** Any comparator (untreated, placebo, conventional treatment) with a control group.

**Outcome.** *Efficacy*. Any quantifiable clinical efficacy score/assessment:

- Clinical symptoms (nose, eye, ear, throat, other related but none specific symptoms e.g., headache, mental function) in the form of symptom scores or improvement/responder rates

- Clinical signs (e.g., peak nasal inspiratory flow (PNIF), nasal cavity examination, computerised tomography (CT) scans)

- Quality of life (QOL) scores (e.g., rhinoconjunctivitis quality of life questionnaire (RQLQ))

- Impairments in daily activities (work, social, professional and educational activities)

- Need for rescue medicine

- Other quantifiable efficacy related scores (e.g., satisfaction score/ effectiveness rating)

   *Safety*. Any adverse event reported.
   Additional exclusion criteria.

- Herbal medicine in the form of mixture

- Allergen immunotherapy

- Combined herbal and procedural treatment (e.g., in combination with surgery or acupuncture) or allergen immunotherapy

- Combined herbal therapy with other complementary treatment (e.g., in combination with probiotics or vitamins)

- Phytochemical compound-based therapy

- External application of herbal medicine (e.g., as herbal patches on acupoints, steam bath etc.)

- Articles that solely reported on serum inflammatory markers without clinical outcomes

- In cross over trials: washout period less than one week (risk of carry-over effect)

- Homeopathy studies

## Search strategy and study selection

Two pairs of independent researchers performed the keyword searches on the electronic databases MEDLINE, CENTRAL, and Web of Science. A predetermined combination of keywords was used consisting of 'allergic rhinitis' AND ('herb*' OR 'phytotherap*' OR 'plant*'). The keyword search was customised to each database. For the database where MeSH terms were not automatically searched (i.e., Web of Science), additional synonyms for allergic rhinitis were searched i.e. "allergic rhinitides" OR "pollen allerg*" OR "pollinos?s" OR "hay fever" OR "hay fever" OR "seasonal allerg*" OR "seasonal rhinitis". The filter for clinical trials or its equivalent, wherever appropriate, was applied. The search was performed for dates since inception of the databases to 18th July 2023. There were no restrictions on language or search duration. An example of the full keyword search strategy can be found in S1 Appendix

   Articles identified through keyword search were managed using a bibliographic manager (EndNote X8.1) to remove duplicates. Two independent researchers performed title, abstract, and full article screening based on agreed inclusion and exclusion criteria while disparities were reviewed by a third. Additional articles were also identified from the reference list of the recent systematic review by Hoang et al [15]. For registered trials listed in our search on clinical trial databases, further efforts were made to identify completed trials and to subsequently include their related publications in this review. For trials with unclear status or trials which were declared as completed but without identifiable publications, we attempted to contact the contact person listed on the trial registers for further clarification.

## Data collection

Two pairs of independent researchers performed full text data extraction using a pre-designed data extraction form (S2 Appendix). Any disagreements were resolved by a third investigator.

All researchers were briefed on the usage of the data extraction form prior to initiating the process to ensure consistency in data extraction.

Key data categories collected included study demographics (author, year, title, country, study objective, study design, trial registry number, ethics approval), population (participant description, inclusion/exclusion criteria, diagnosis and subtype, age, sex, sample size, drop outs, sample size calculation and power of the study), intervention (plant name, part, form, formulation and its contents, dose and duration, quality data of investigational product e.g., source, qualitative/quantitative analysis, voucher specimen), comparator (description, dose and duration), outcomes (efficacy and safety), and others (medication adherence, funding).

All relevant clinical efficacy outcomes (Table 1), including symptoms, quality of life, impairments to productivity, and patients' own effectiveness/satisfaction scores and ratings were considered to be of significance for exploration in this review. This is in line with the ARIA guidelines' recommendations that both symptom control and patients' preference as well as empowerment are important considerations in allergic rhinitis management [5]. For efficacy outcomes, the sample size of each intervention/comparator arm, mean and standard deviation (SD) were sought as the preferred data. If unavailable, however, standard error (SE), p values, and 95% confidence intervals (CI) were collected. For dichotomous outcomes, data on the actual number of events were preferably sought. Alternatively, the percentage of events were noted instead. For all outcomes, the baseline and post intervention values and/or changes in values after intervention measured at all time points were collected. In cases of studies which did not report on any of these numerical values or if there were ambiguities, attempts to contact the authors for clarifications were made. If the team received no responses for the numerical values of these data, the team estimated the respective values from figures using WebPlotDigitizer version 4.5 [17].

**Table 1. Definition of efficacy and safety outcomes.**

| Outcome | Definition |
|---|---|
| Clinical symptoms | Nose, eye, ear, throat symptoms scores, or global assessment scores (e.g., TNSS, SNOT-22, RCAT)<br>None allergic rhinitis specific symptom scores (e.g., headache, mental function, sleep impairment)<br>Responder/improvement rates (i.e., number/percentage of patients who reported improvement in symptoms) |
| Clinical signs | Physical examination related parameters (e.g., PNIF, CT scan score, nasal cavity measurements) |
| QOL | Scales or scores assessing the quality of life (e.g., RQLQ, SF-36) |
| Activity impairment | Scales or scores assessing the impairment in day-to-day functions or activities (e.g., WPAI) |
| Rescue medicine | Measurement on the need for rescue medicine (e.g., antihistamines) on top of treatment (e.g., number of doses, number of patients, medication scores) |
| Others | Effectiveness scores or satisfaction scores rated by patients |
| Safety | Adverse events (descriptive and numerical) |

Computerised tomography (CT), Peak nasal inspiratory flow (PNIF), Quality of life (QOL), Rhinitis control assessment test (RCAT), Rhinoconjunctivitis quality of life questionnaire (RQLQ), Sino-nasal outcome test (SNOT-22), 36-Item short form survey (SF-36), Total nasal symptom score (TNSS), Work productivity and activity impairment questionnaire (WPAI)

## Data analysis

Two pairs of independent researchers performed risk of bias assessment and analyses. A third investigator was consulted in the case of disagreements.

**Qualitative analysis.** All included studies were qualitatively and numerically analysed for study characteristics on study design, sample size, sex, diagnosis and subtype, intervention and comparator details, and category of study outcomes. Safety reports of medicinal plants for allergic rhinitis were also descriptively analysed based on reported adverse events.

**Risk of bias assessment.** Risk of bias assessments were carried out on the following domains of sequence generation, allocation concealment, blinding of patient and personnel, blinding of outcome assessors, incomplete outcome data, selective outcome reporting, and other biases. Other biases included consideration of industrial-sponsored or conducted trials, as well as additional bias sources for randomised cross-over trials. These include risk of carry over effects and availability of first-period data, as recommended in Section 23.2.3 of the Cochrane Handbook for Systematic Reviews of Interventions [18]. Risk of bias was judged as 'High', 'Low', or 'Unclear' for each domain of each trial. In general, if the required information to reasonably assess risk of bias was not clearly reported, a score of 'Unclear' was assigned. Otherwise, it was judged as either 'High' or 'Low' based on clearly reported methods. The risk of bias graph and summary was generated using RevMan 5.4 [19].

**Article reporting quality.** Two pairs of independent researchers performed assessment on the reporting quality of the medicinal plant-based investigational product in each trial, based on the criteria stated in the CONSORT extension for reporting herbal medicines item No.4 [20, 21].

**Meta-analyses.** Meta-analyses and the generation of forest plots were carried out following the recommended methods of the Cochrane Handbook for Systematic Reviews of Intervention using RevMan 5.4 [18, 19]. Pairwise meta-analyses were conducted for all quantifiable outcomes to generate pooled outcome estimates. In cases where there were distinctive differences in study design (e.g., intervention used for symptom relief instead of control, or where intervention was not uniformly applied) was identified, narrative analysis was performed instead. For multi-arm studies, treatment arms of different doses were pooled as a single group and compared to another single comparator group if necessary.

Pooled outcome estimates were generated using the random effects model for continuous data were reported as mean difference (MD) or standardised mean difference (SMD) with 95% CI. Pooled outcome estimates for dichotomous outcomes were generated using the Mantel-Haenszel methods and reported as risk ratios (RRs). Further details on data processing during analyses can be found in S3 Appendix.

*Assessment of heterogeneity & subgroup analyses.* The $I^2$ statistic was used to assess the heterogeneity of included studies. A cut-off of 50% was set and meta-analyses with values above this were considered as having substantial heterogeneity in which the exploration of possible explanations were justified, as recommended in Section 10.10.1 the Cochrane Handbook for Systematic Reviews of Intervention [18]. Possible factors that were explored include differences in the intervention (selection of medicinal plants), dose, duration, participant characteristics, study design, and risk of bias. The team carried out sub-group analyses based on the identified potential reasons if there were sufficient studies available in each sub-group.

*Sensitivity analysis.* In the event studies with high risk of bias in individual meta-analyses were included, the team would have performed sensitivity analyses to meaningfully assess the impact of excluding such studies on the pooled estimates, if sufficient number of studies were available. However, as the number of included studies was insufficient, this was not performed.

*Reporting bias and missing data.* For individual studies, the review team assessed reporting bias based on the selective outcome reporting and incomplete outcome reporting domain of the risk of bias assessment mentioned earlier.

For assessment on overall publication bias, the team would have used a funnel plot to assess reporting biases for outcomes with more than 10 included studies within the same meta-analysis. This was not performed as there was insufficient number of included studies for each outcome. A narrative analysis on the proportion of publicly available publications compared to registered trials was performed instead.

**Certainty of evidence.** A pair of researchers performed ratings on the certainty of evidence using the GRADE approach for each of the outcomes that contributed data to the meta-analyses in this review, based on 5 GRADE criteria (study limitations, consistency of effect, imprecision, indirectness, and publication bias) Table 14.1.a of the Cochrane Handbook for Systematic Reviews of Intervention [18] was used to guide the assessment of individual and overall risk of bias across the pooled studies. The assessment was checked by a senior methodologist. If any issues were identified to pose a serious risk to affect the outcome estimate, the review team downgraded the certainty of evidence by one level. If the issue was considered to be very serious, the team downgraded the evidence by two levels. For studies with high heterogeneity with $I^2$ of greater or equals to 90%, the evidence is downgraded by two levels. A summary of findings table on all outcomes, level of evidence, and justification of rating decisions was generated using GRADEpro online platform [22].

## Protocol registration

This review is registered with PROSPERO (CRD42021279723).

## Results

A total of 29 published randomised controlled trials (26 parallel, 3 cross-over, participants = 1879) were included out of 1523 articles retrieved from the initial search. Of the 29 published trials, 27 (n = 1769) contributed suitable data for quantitative analysis, as outlined in the PRISMA flow diagram (Fig 1). A list of excluded studies after full text reading with reasons for exclusion is included in S4 Appendix.

## Study characteristics

The main characteristics of the 29 included published articles are presented in S1-S3 Tables in S5 Appendix according to their PICO elements. Almost all [23–50] were conducted in adult populations while only one study [51] investigated the effects of medicinal plants among children. Of the 29 included studies, more than half reported a study duration of 28 days or longer. One study [43] did not apply the same treatment duration across participants and was therefore not included for quantitative analysis. Oral route was the main route of administration (82.8%) while the remainder were given intranasally.

For the types of medicinal plants studied, *Petasites hybridus* was the most frequently investigated (5 studies), followed by *Urtica dioica* L. and *Cinnamomum zeylanicum* (2 studies each). The other articles reported on the efficacy of a variety of medicinal plants. Rescue medications such as antihistamines were allowed in 13.8% of the studies. 28 studies [23–34, 36–51] investigated medicinal plants for symptomatic control (i.e., daily administration) while one study [35] used the medicinal plant as rescue therapy (i.e., only on symptom onset) which was therefore not included for quantitative analysis.

Nose and ear symptoms were the most frequently reported clinical outcomes when investigating the efficacy of medicinal plants for allergic rhinitis. These can be reported as overall

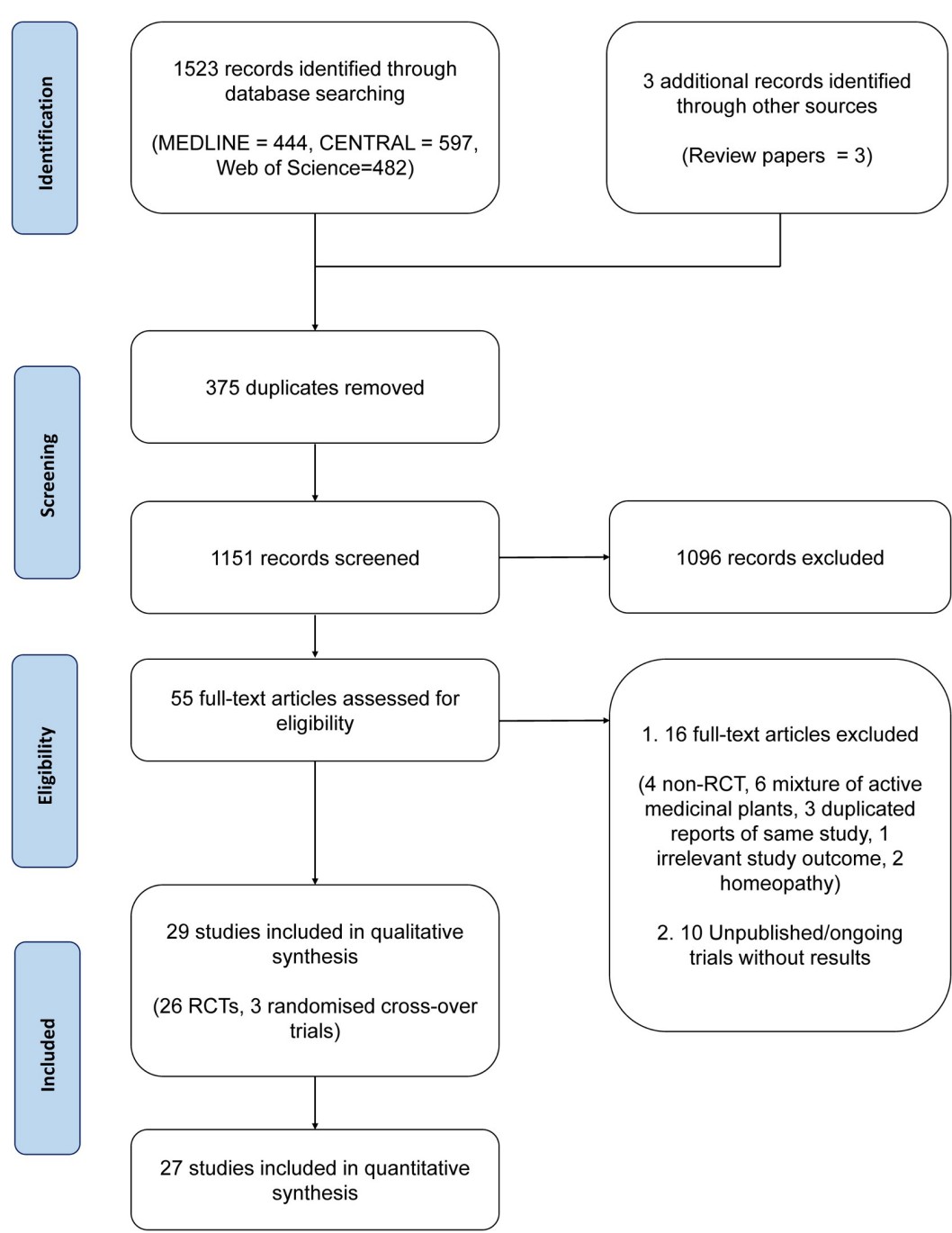

**Fig 1. PRISMA flow diagram of included studies.** Randomised controlled trial (RCT).

total symptom scores, individual symptom scores, or based on global assessment of overall improvement by patients and/or attending physicians. The RQLQ, including its variations such as the mini-RQLQ and the Japanese-RQLQ was the more commonly used validated questionnaire in assessing the quality of life and impact of allergic rhinitis among study subjects.

### Unpublished trial data

In addition to the 29 published trials included for analysis, we identified 10 registered trials that are yet unpublished or ongoing. Among these, three were completed studies without published data and seven are still ongoing. The details of these trials are presented in S6 Appendix.

### Risk of bias

Of the 29 included studies, half (51.7%) were deemed to have low risk of selection bias with clear reporting of method used for random sequence generation. Allocation concealment was not clearly described in more than half (58.6%) of the studies included. Performance bias was judged as high in three studies (10.3%) whereby Derakshan et al. administered intervention and control in two different dosage forms (tablet and powder) [29], while Wu et al. [51] administered nasal drops in the treatment group and tablets in the control group, and Atar et al. [50] allocated patients with different form of nasal sprays and irrigation devices. Risk of detection bias was considered as high in one study [29] whereby the researchers were unable to be blinded due to distinctive differences in the dosage forms given to the treatment and control arms. Five studies (17.2%) [23, 26, 28, 35, 39] were considered as having high risk of attrition bias. Imbalances in sample size or drop out-rates between the treatment and control arms [23, 26, 35, 39] and high participant drop-out rates (>20%) [26, 35] was observed in four and two studies respectively. Reporting bias was considered to be high in six studies (20.7%) [28, 35, 39, 42, 43, 46] where these studies did not report baseline data of the participants and/or relevant clinical outcome(s) of allergic rhinitis. Other biases were considered to be unclear in 16 studies (55.2%) [25, 27, 28, 31, 34–43, 46, 48] due to unclear funding source or were industry-sponsored, conducted, or authored trials. The remaining three with unclear risk for other biases were cross-over studies. These three studies [30, 33, 49] did not reported clearly on the cross over methods, justification of wash over period, or there was no separate reporting of data from each phase of the study. The overall risk of bias summary and its details for individual studies are shown in Figs 2 and 3 where green, yellow, and red represents low, unclear, and high risk of bias respectively in each domain.

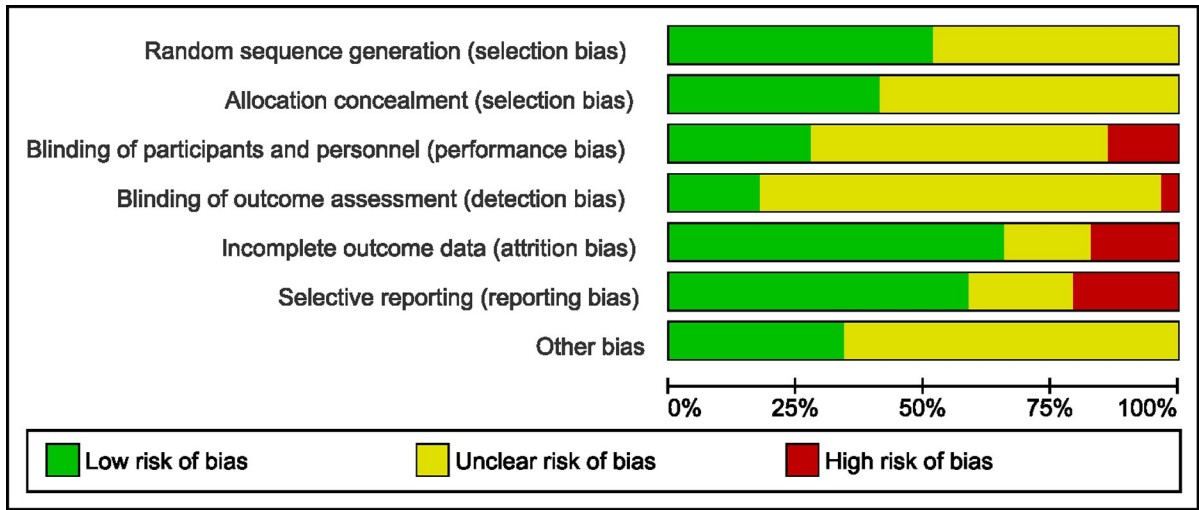

**Fig 2. Risk of bias summary of included studies.**

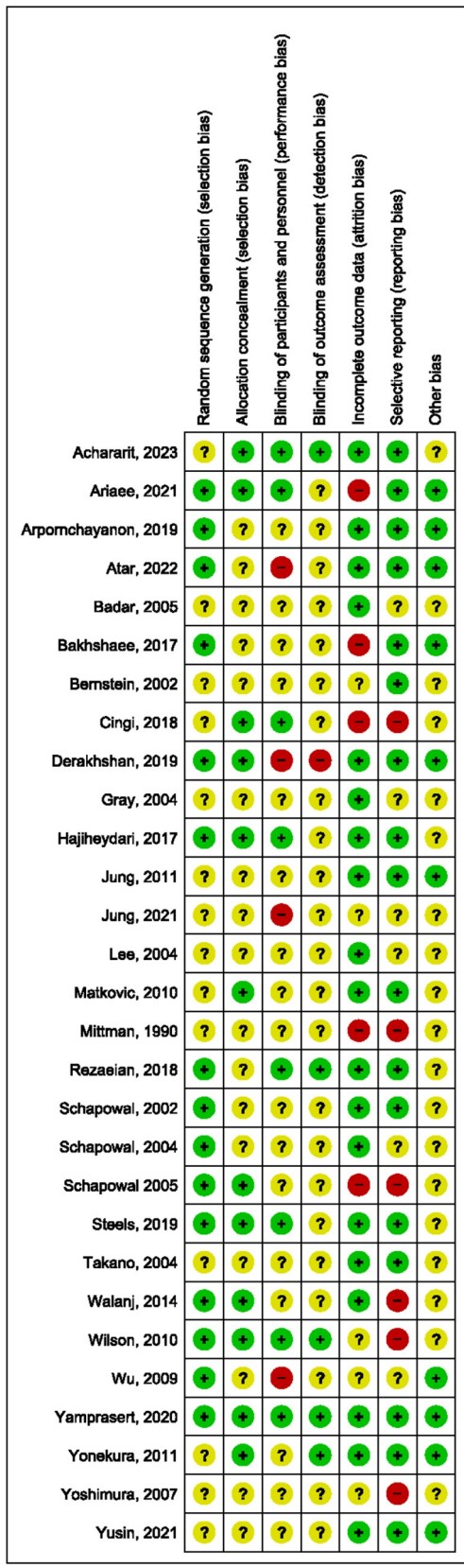

**Fig 3. Detailed risk of bias analysis of individual studies.**

## Effects of interventions

Meta-analyses on efficacy outcomes were conducted according to four major comparisons major groups (1) medicinal plant vs placebo, (2) medicinal plant vs antihistamine, (3) medicinal plant vs intranasal corticosteroids, and (4) medicinal plant as add-on to conventional therapy vs conventional therapy only. Details on adverse events are presented in a separate subsection titled 'safety'.

**Medicinal plant vs placebo.**   Overall, 20 studies [25, 27, 28, 30–35, 38–43, 45–49] compared a medicinal plant against placebo for allergic rhinitis.

*(a) Nasal and eye symptoms score (Post treatment mean).* A graphical representation of these findings is presented in Fig 4.

**Total nasal and eye symptoms**. Pooled estimates from two studies [27, 40] showed no differences in total nasal and eye symptom scores between the medicinal plant-treated (55 participants) and placebo arms (56 participants) (SMD -0.45, 95% CI -1.53 to 0.63; participants = 111; studies = 2; $I^2$ = 87%) but the evidence is very uncertain. There is substantial heterogeneity, as shown by the $I^2$ of 87%. One study [40] administered intranasal *C. zeylanicum* for 7 days while the other [27] administered oral grapeseed extract for 56 days, both involving seasonal allergic rhinitis adult patients. There were insufficient studies for subgroup analysis.

One study [43] reported similar improvements in total nasal and eye symptom scores among those treated with standardised bark extract of the French maritime pine (*Pinus pinaster* Ait.) and placebo over the entire birch pollen season. This study applied different durations of treatment (3–8 weeks) across the participants and it was reported that the best improvements in symptom scores were among those given treatment 7–8 weeks prior to the allergy season. This study was not included in the meta-analysis.

**Total night-time nasal and eye symptoms**. Results from one study [40] suggested at improvements in total night time nasal and eye symptom scores for those treated with the medicinal plant (MD -1.37, 95% CI -2.41 to -0.33; participants = 60; studies = 1).

**Total nasal symptoms**. Pooled estimates from five studies [30, 32, 33, 42, 49] suggested small improvements in total nasal symptoms scores between the medicinal plant-treated (126 participants) and placebo arms (123 participants) (SMD -0.31, 95% CI -0.59 to -0.02; participants = 249; studies = 5; $I^2$ = 21%).

**Total day-time nasal symptoms**. Results from one study [40] suggested that use of the medicinal plant likely results in an improvement in total day-time nasal symptoms (MD -5.47, 95% CI -7.85 to -3.09; participants = 60; studies = 1).

**Rhinorrhoea**. Pooled estimates from six studies [27, 28, 30, 31, 42, 48] showed that medicinal plants may have no effect on rhinorrhoea (SMD -0.85, 95% CI -1.73 to 0.04; participants = 399; studies = 6; $I^2$ = 94%). The evidence is very uncertain. There is substantial heterogeneity, as shown by the $I^2$ of 94%. All six studies investigated different medicinal plants while one [42] of the six administered the treatment intranasally (the rest through oral ingestion). Four studies [30, 31, 42, 48] had a shorter study duration of 14–28 days, while the remaining two conducted the study for longer periods of 56 [27] and 147 [28] days. We explored plausible factors that might account for the heterogeneity observed in terms of population, intervention, comparison and outcomes, and did not find any that substantially affect the $I^2$.

**Nasal congestion**. Pooled estimates from seven studies [27, 28, 30, 31, 38, 42, 48] suggested that medicinal plants may improve nasal congestion but the evidence is very uncertain (SMD

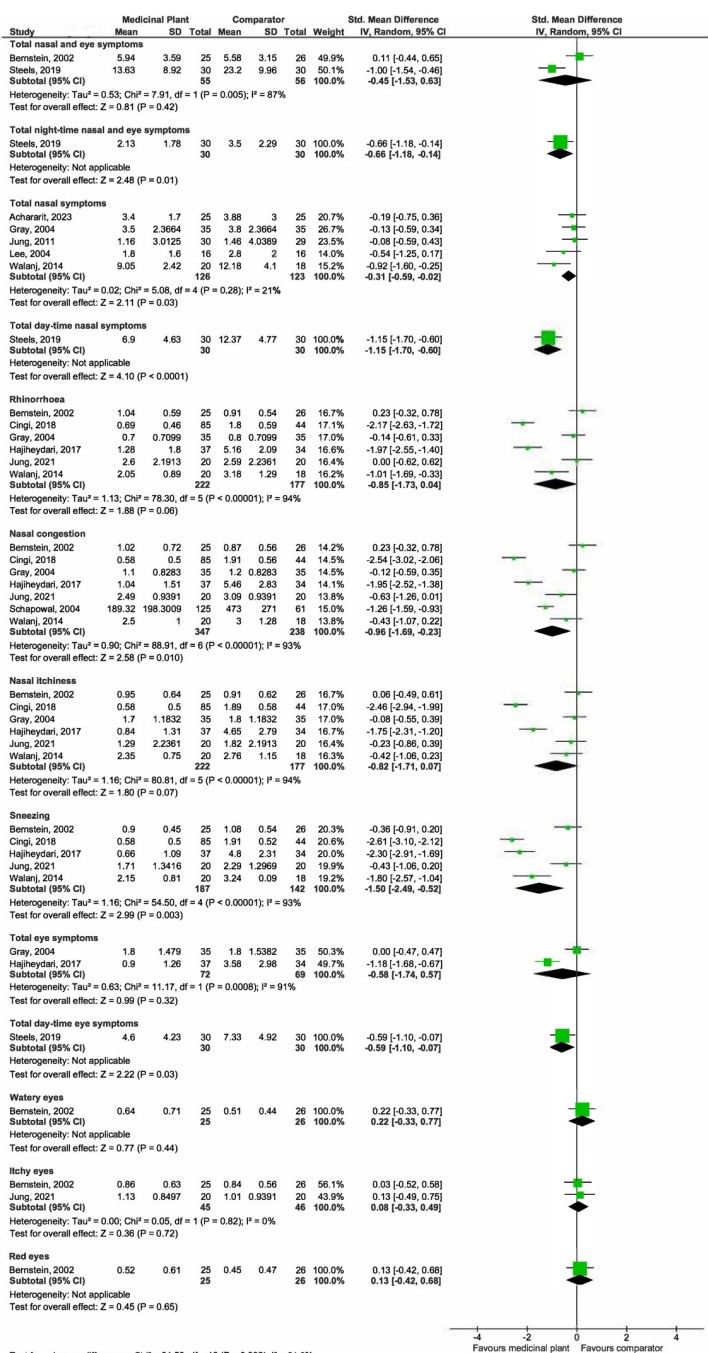

**Fig 4. Forest plot comparison of medicinal plant vs placebo for nasal and eye symptoms scores (post treatment mean).**

-0.96, 95% CI -1.69 to -0.23; participants = 585; studies = 7; $I^2$ = 93%). There is substantial heterogeneity, as shown by the $I^2$ of 93%. Two studies [30, 38] investigated the same plant *P. hybridus* though there were insufficient details reported to compare the formulations while the remaining four investigated different medicinal plants. One [42] out of seven administered the treatment intranasally (the rest through oral ingestion). Five studies [30, 31, 38, 42, 48] had a shorter study duration of 14–28 days, while the remaining two conducted the study for longer

periods of 56 [27] and 147 [28] days. We explored plausible factors that might account for the heterogeneity observed in terms of population, intervention, comparison, and outcomes, and did not find any that substantially affect the $I^2$. Sub-group analysis by plant type also did not substantially reduce the $I^2$ value.

***Nasal itchiness***. Pooled estimates from six studies [27, 28, 30, 31, 42, 48] showed that medicinal plants may have no effect on nasal itchiness scores but the evidence is very uncertain (SMD -0.82, 95% CI -1.17 to 0.07; participants = 399; studies = 5; $I^2$ = 94%). There is substantial heterogeneity, as shown by the $I^2$ of 94%. All six studies investigated different medicinal plants while one [42] out of six administered the treatment intranasally (the rest through oral ingestion). Four studies [30, 31, 42, 48] had a shorter study duration of 14–28 days, while the remaining two conducted the study for longer periods of 56 [27] and 147 [28] days. We explored plausible factors that might account for the heterogeneity observed in terms of population, intervention, comparison, and outcomes, and did not find any that substantially affect the $I^2$.

***Sneezing***. Pooled estimates from five studies [27, 28, 31, 42, 48] showed that those treated with medicinal plants reported significantly less sneezing (187 participants) than those with placebo (142 participants) (SMD -1.50, 95% CI -2.49 to -0.52; participants = 329; studies = 5; $I^2$ = 93%). However, the evidence is very uncertain. There is substantial heterogeneity, as shown by the $I^2$ of 93%. All studies investigated different medicinal plants. One [42] out of five administered the treatment intranasally (the rest through oral ingestion). Three studies [31, 42, 48] had a shorter study duration of 28 days, while the remaining two conducted the study for longer periods of 56 [27] and 147 [28] days. We explored plausible factors that might account for the heterogeneity observed in terms of population, intervention, comparison, and outcomes, and did not find any that substantially affect the $I^2$.

***Total eye symptoms***. Pooled estimates from two studies [30, 31] suggested no differences in total eye symptom scores between the medicinal plant-treated (72 participants) and placebo arms (69 participants) (SMD -0.58, 95% CI -1.74 to 0.57; participants = 141; studies = 5; $I^2$ = 91%) though the evidence is very uncertain. There is substantial heterogeneity, as shown by the $I^2$ of 95%. Both studies investigated oral formulations of different medicinal plants for a duration of 14 [30] and 28 [31] days respectively. There were insufficient studies for subgroup analysis.

***Total day-time eye symptoms***. Results from one study [40] suggested that medicinal plant use likely results in a slight improvement in total day-time eye symptom scores (MD -2.73, 95% CI -5.05 to -0.41; participants = 60; studies = 1).

***Watery, itchy, and red eyes***. Results from a single study [27] suggested that medicinal plants may not improve watery and red eyes (Watery eyes: MD 0.13, 95% CI -0.20 to 0.46; Itchy eyes: MD 0.02, 95% CI -0.31 to 0.35; Red eyes: MD 0.07, 95% CI -0.23 to 0.37; participants = 51; studies = 1).

Pooled results from two studies [27, 48] suggested that medicinal plants may not improve eye itchiness (SMD 0.08, 95% CI -0.33 to 0.49; participants = 91; studies = 2).

*(b) Nasal, eye, and throat symptoms score (Changes in mean)*. The graphical representation of these findings is presented in Fig 5.

***Total nasal, eye, and throat symptoms (reflexive & instantaneous)***. Results from a single study [39] suggested that medicinal plant likely results in an improvement in total reflexive and instantaneous nasal, eye, and throat symptoms scores (Reflexive: MD -3.46, 95% CI -4.33 to -2.59; Instantaneous: MD -8.30, 95% CI -10.64 to -5.96; participants = 217; studies = 1).

***Total nasal symptoms***. Pooled estimates from three studies [34, 46, 47] showed that those treated with medicinal plants may have lesser total nasal symptoms (62 participants) compared

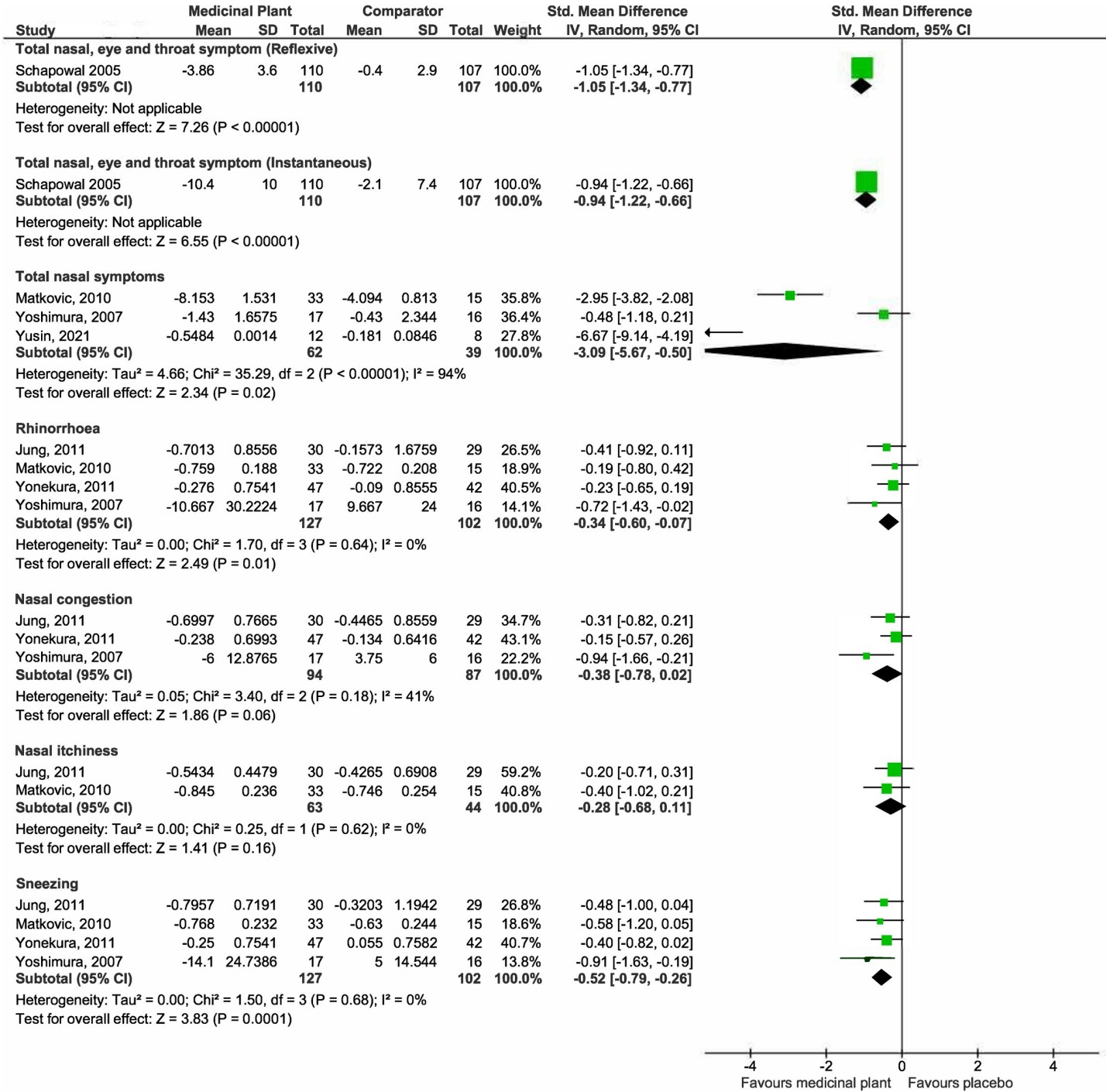

**Fig 5. Forest plot comparison of medicinal plant vs placebo for nasal, eye, and throat symptoms scores (changes in mean).**

to those treated with placebo (39 participants) (SMD -3.09, 95% CI -5.67 to -0.50; participants = 101; studies = 3; $I^2$ = 94%) but the evidence is very uncertain. There is substantial heterogeneity, as shown by the $I^2$ of 94%. All studies investigated different medicinal plants via the oral route, with study durations ranging from 21 to 56 days. We explored plausible factors

that might account for the heterogeneity observed in terms of population, intervention, comparison, and outcomes, and did not find any that substantially affect the $I^2$.

*Rhinorrhoea*. Pooled estimates from four studies [32, 34, 45, 46] showed that medicinal plant use may result in slight improvement in rhinorrhoea (SMD -0.34, 95% CI -0.60 to -0.07; participants = 229; studies = 4; $I^2$ = 0%).

*Nasal congestion*. Pooled estimates from three studies [32, 45, 46] suggested that medicinal plants may result in little to no difference in nasal congestion scores (SMD -0.38, 95% CI -0.78 to 0.02; participants = 181; studies = 3; $I^2$ = 41%).

*Nasal itchiness*. Pooled estimates from two studies [32, 34] suggested that medicinal plants may result in little to no difference in nasal itchiness scores (SMD -0.28, 95% CI -0.68 to 0.11; participants = 107; studies = 2; $I^2$ = 0%).

*Sneezing*. Pooled estimates from four studies [32, 34, 45, 46] suggested that medicinal plants result in less sneezing (SMD -0.52, 95% CI -0.79 to -0.26; participants = 229; studies = 4; $I^2$ = 0%).

*(c) Global assessment score*. Pooled estimates from two studies [34, 37] showed no significant differences in global assessment scores between the medicinal plant-treated (158 participants) and placebo arms (78 participants) (SMD -3.06, 95% CI -7.26 to 1.14; participants = 236; studies = 2; $I^2$ = 98%) but the evidence is very uncertain (Fig 6). There is substantial heterogeneity, as shown by the $I^2$ of 98%. Both studies investigated oral formulations of different medicinal plants with study durations of 14 [38] and 42 [34] days respectively. There were insufficient studies for subgroup analysis.

*(d) Responder rates*.

*Global symptoms improvement*. Pooled estimates from five studies [27, 38, 39, 41, 45] showed that those treated with medicinal plants were more likely to experience overall symptomatic improvement (251 participants) than of placebo (255 participants) (RR 1.43, 95% CI 1.09 to 1.87; participants = 506; studies = 5; $I^2$ = 56%) (Fig 7). There is substantial heterogeneity, as shown by the $I^2$ of 56%. All five studies investigated orally administered formulations. Two studies [38, 39] investigated the same plant *P. hybridus*. In both studies, responder rates were collected as secondary outcomes and defined as improvement in more than 25%-50% of the main total symptom scores using a visual analogue score. Two studies allowed the use of rescue medications (antihistamines) on top of treatment [27, 45]. The study duration in all studies varied from 21 to 56 days. We explored plausible factors that might account for the heterogeneity observed in terms of population, intervention, comparison, and outcomes, and found that pooled estimates from two studies [27, 45] which allowed the use of rescue medications were more homogenous but suggested that medicinal plants result in little to no difference in responder rates (RR 1.09, 95% CI 0.72 to 1.64; participants = 140; studies = 2; $I^2$ = 0%). The three studies [38, 39, 41] which did not allow the use of additional rescue medication

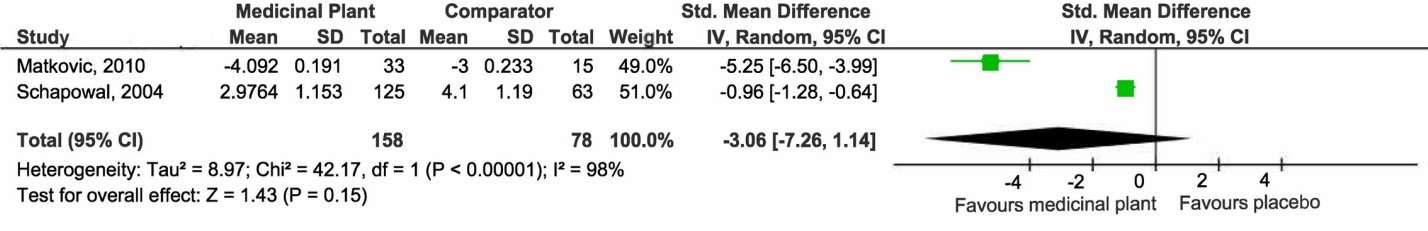

**Fig 6. Forest plot comparison of medicinal plant vs placebo for global symptoms assessment score.**

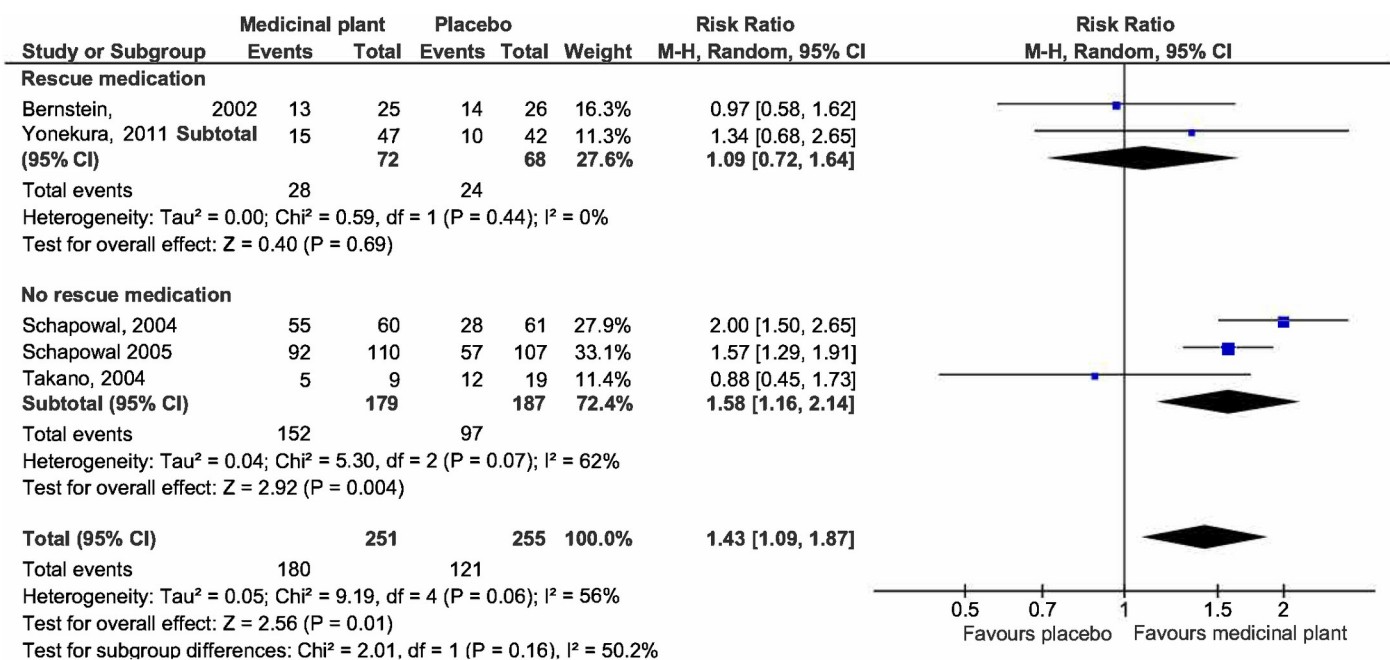

**Fig 7. Forest plot comparison of medicinal plant vs placebo for responder rates with global symptoms improvement with subgroup analysis by allowed rescue medications.**

suggested at improved responder rates but they still had substantial heterogeneity (RR 1.58, 95% CI 1.16 to 2.14; participants = 366; studies = 3; $I^2$ = 62%). Additional subgroup analysis of these three studies which did not allow the use of additional rescue medication by plant type did not further reduce the $I^2$ substantially ($I^2$ = 47%).

*Nasal symptoms*. Two studies [25, 41] reported responder rates with improved individual nasal symptoms of rhinorrhoea, nasal congestion, nasal itchiness, and sneezing (Fig 8). Participants treated with medicinal plants were more likely to experience an improvement in rhinorrhoea (RR 5.66, 95% CI 2.67 to 11.99; participants = 97; studies = 2; $I^2$ = 0%) and nasal itchiness (RR 7.92, 95% CI 2.93 to 21.38; participants = 82; studies = 2; $I^2$ = 0%). There is substantial heterogeneity in the outcomes for nasal congestion and sneezing which did not show significant differences in the likelihood of reporting improvement between the treated and untreated groups. There were insufficient studies for subgroup analysis.

*Eye symptoms*. Results from one study [41] that reported on responder rates for watery and itchy eyes symptom was excluded from analysis as there were no events in the comparator arm (S1 Fig in S7 Appendix).

*(e) Symptom duration score*. One study [32] suggested at no improvements in symptom duration scores with medicinal plant (MD -0.20, 95% CI -2.03 to 1.63; participants = 59; studies = 1) (S2 Fig in S7 Appendix).

*(f) Clinical signs*.

*Peak nasal inspiratory flow (PNIF)*. Two studies [47, 48] suggested no improvements in PNIF among those treated with medicinal plants compared to placebo. These two studies could not be pooled due to the differences in units and methods of measurements.

*(g) Quality of life (QOL)*.

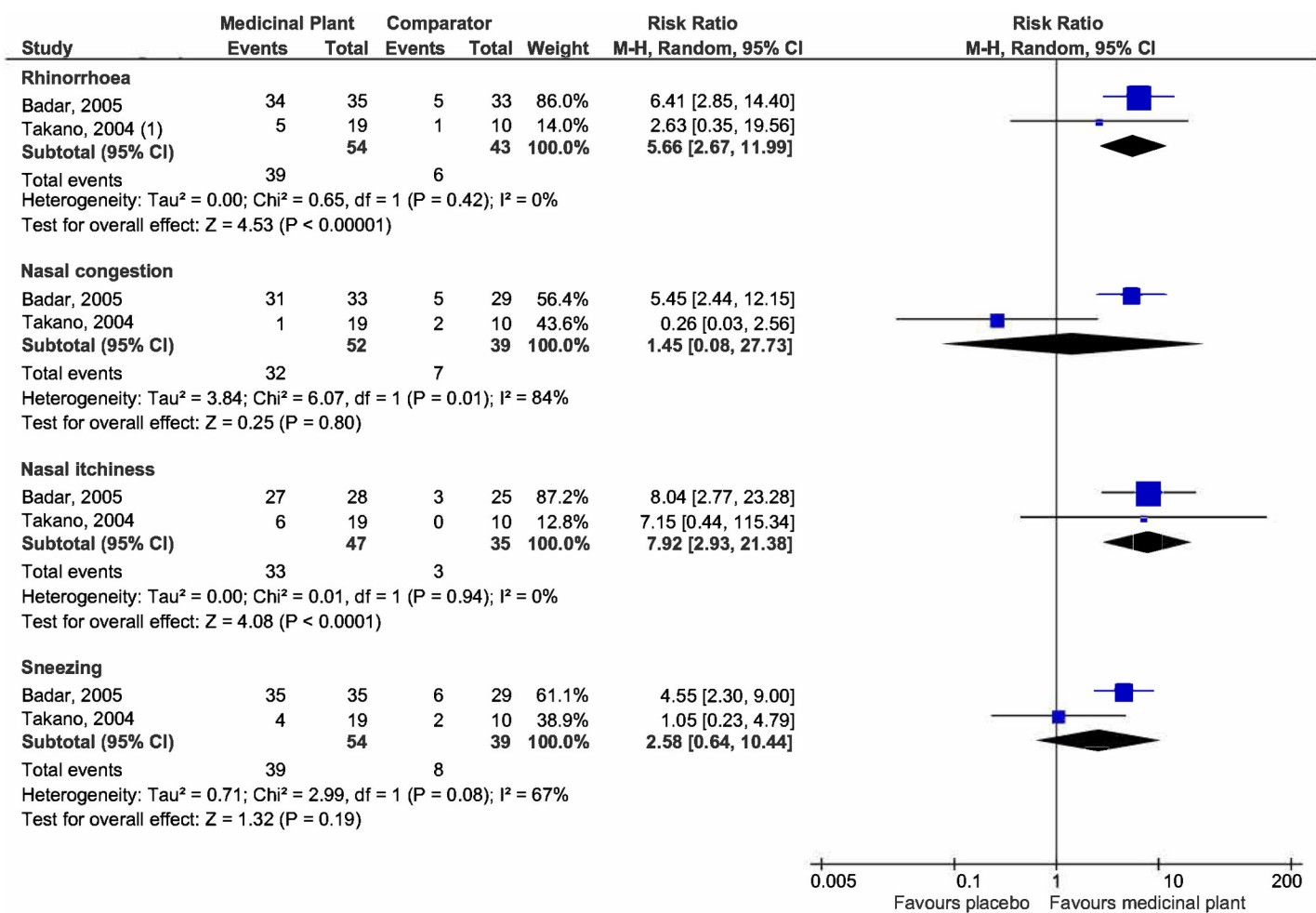

**Fig 8. Forest plot comparison of medicinal plant vs placebo for responder rates (nasal symptoms).**

***Rhinoconjunctivitis Quality of Life Questionnaire (RQLQ).*** Four studies [27, 32, 40, 42] reported total RQLQ scores while three studies [27, 40, 42] presented the scores of its individual components. There is substantial heterogeneity in all pooled outcomes (S1 Fig in S8 Appendix). By excluding the one study with a short duration of study (7 days) [40] while the remaining studies had 28 days or longer study duration, the $I^2$ was substantially improved. Pooled estimates from the remaining three studies suggested that medicinal plants may result in slight improvements in total RQLQ scores, activity limitations, nose symptoms, and eye symptoms but not sleep problems, non-nose/eye symptoms, practical problems, and emotional function (Fig 9).

***Mini-RQLQ.*** One study [30] suggested that the medicinal plant improved total mini-RQLQ scores (Total: MD -2.02, 95% CI -2.64 to -1.39; participants = 48; studies = 1). Another single study [30] suggested no improvements in individual components of mini-RQLQ scores (Activity limitations: MD -0.20, 95% CI -0.75 to 0.35; Practical problems: MD -0.20, 95% CI -0.87 to 0.47; Nose symptoms: MD -0.50, 95% CI -1.11 to 0.11; Other symptoms: MD -0.10, 95% CI -0.81 to 0.61; participants = 70; studies = 1). All results are presented in S3 Fig in S7 Appendix.

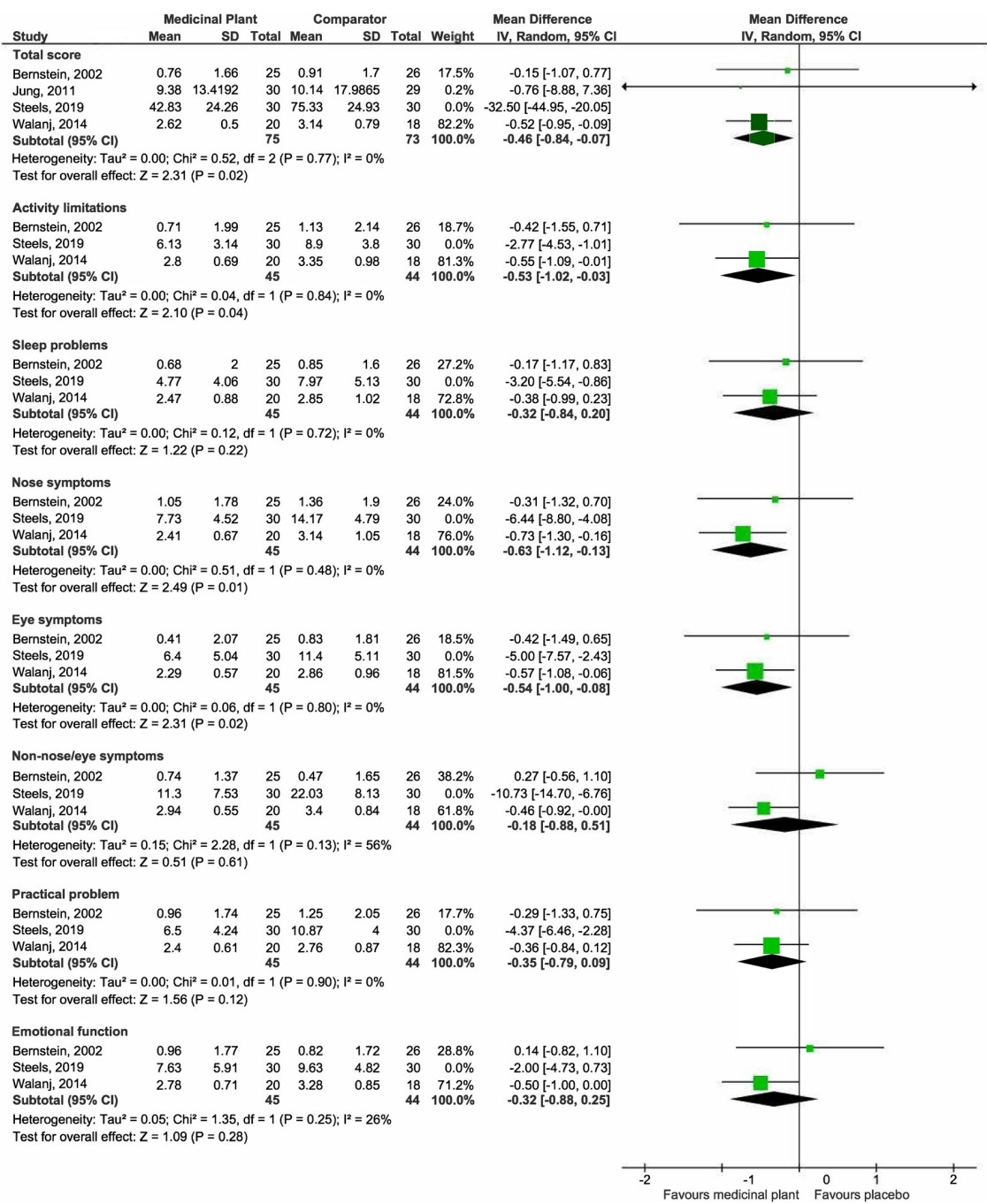

**Fig 9. Forest plot comparison of medicinal plant vs placebo for RQLQ.**

*Other QOL scores*. One study [46] suggested that medicinal plant may improve other QOL scores (unspecified scale) (MD -8.50, 95% CI -14.75 to -2.25; participants = 33; studies = 1) (S4 Fig in S7 Appendix).

*(h) Impairment in daily activities.*

A single study [40] evaluated suggested that the medicinal plant probably does not reduce the working hours missed due to allergy and may have no impact on productivity impairments. Pooled estimates from two studies suggested that medicinal plants may not improve regular day activity (SMD -0.57, 95% CI -1.28 to 0.14; participants = 81; studies = 2; $I^2$ = 60%) (Fig 10). There is substantial heterogeneity, as shown by the $I^2$ of 60%. Both studies investigated intranasal preparations of the same plant *C. zeylanicum* for 7 [40] and 28 [42] days. There were insufficient studies for subgroup analysis.

*(i) Need for rescue medicine.*

One study suggested at no difference in medication use scores [45] among those treated with the medicinal plant and placebo (MD -0.41, 95% CI -1.33 to 0.51; participants = 89; studies = 1) while another [27] suggested at no difference in likelihood of needing rescue medications (RR 1.32, 95% CI 0.75 to 2.34; participants = 51; studies = 1) (S5 and S6 Figs in S7 Appendix).

*(j) Effectiveness/satisfaction score.*

One study [28] suggested that the medicinal plant probably improves intervention effectiveness and patient satisfaction scores (Effectiveness score: MD -3.90, 95% CI -4.35 to -3.45; Satisfaction score: MD -3.81, 95% CI -4.36 to -3.26; participants = 129; studies = 1) (S7 and S8 Figs in S7 Appendix).

One study [35] investigated the use of freeze-dried *U. dioica* capsule on an as needed basis during the onset of symptoms. Excluding patients with mild symptoms, half of the patients felt that the medicinal plant capsules improved or maintained their symptoms and would purchase them for future use while the other half had worsened symptoms and would not purchase them. Approximately 15% of participants who took the placebo capsules felt that they would purchase them for future use. This study was not included for quantitative analysis.

**Medicinal plant vs antihistamine.** Overall, eight studies [29, 33, 37, 39, 44, 47, 48, 51] compared medicinal plants against antihistamines for allergic rhinitis.

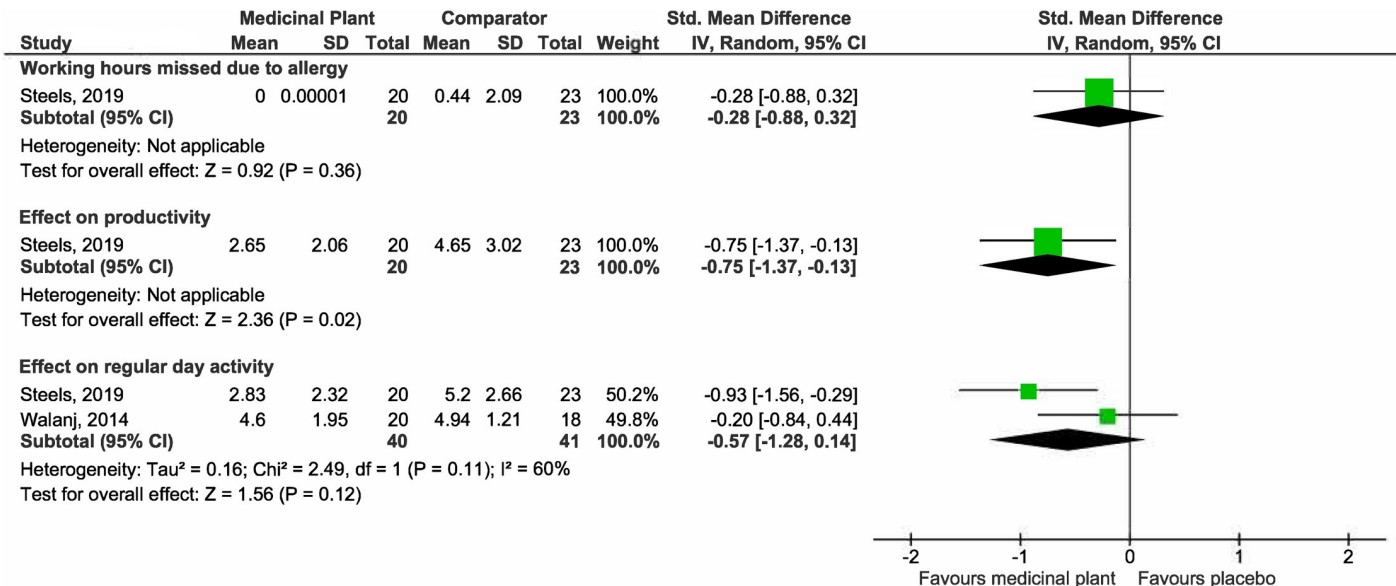

**Fig 10. Forest plot comparison of medicinal plant vs placebo for impairment in daily activity.**

*(a) Nasal and eye symptom score (post treatment mean).* The graphical representation of these findings is presented in Fig 11.

***Total nasal symptoms***. Pooled estimates from two studies [29, 44] suggested at no significant differences in total nasal symptoms scores between the medicinal plant-treated (74

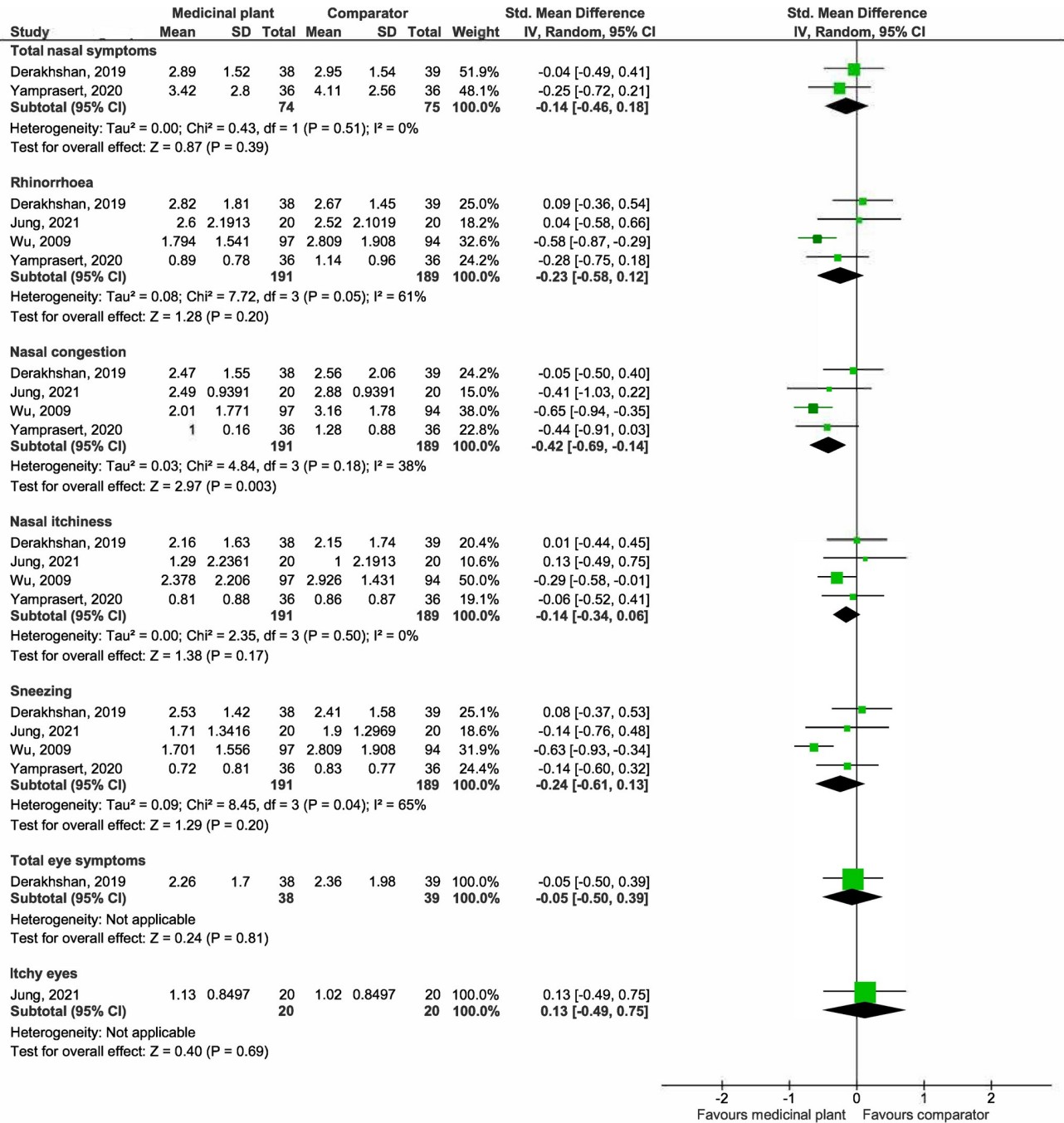

**Fig 11. Forest plot comparison of medicinal plant vs antihistamine for nasal and eye symptoms scores (post treatment mean).**

participants) and placebo arms (75 participants) (SMD -0.14, 95% CI -0.46 to 0.18; partici-
pants = 149; studies = 2; $I^2$ = 0%).

*Rhinorrhoea*. Pooled estimates from four studies [29, 44, 48, 51] showed no significant dif-
ferences in rhinorrhoea scores between the medicinal plant-treated (191 participants) and pla-
cebo arms (189 participants) (SMD -0.23, 95% CI -0.58 to 0.12; participants = 380; studies = 4;
$I^2$ = 61%). There is substantial heterogeneity, as shown by the $I^2$ of 61%. One study was con-
ducted among children [51] using an intranasal formulation, while the remaining three [29,
44, 48] investigated oral formulations of different medicinal plants in adults. The duration of
study varies from 14 to 42 days. We explored plausible factors that might account for the het-
erogeneity observed in terms of population, intervention, comparison, and outcomes, and
found that by excluding one study [51] on children, the $I^2$ reduced to 0% with pooled estimates
showing no significant difference between treatment groups (SMD -0.06, 95% CI -0.35 to 0.22;
participants = 189; studies = 3; $I^2$ = 0%). Co-incidentally, this also reflects subgroup analysis by
route of administration (i.e., children with intranasal administration vs adults with oral admin-
istration). The remaining single study suggested that medicinal plant may slightly improve rhi-
norrhoea among children (SMD -0.58, 95% CI -0.87 to -0.29; participants = 191; studies = 1)
(S1 Fig in S9 Appendix).

*Nasal congestion*. Pooled estimates from four studies [29, 44, 48, 51] showed that those
treated with medicinal plants reported significantly less nasal congestion (191 participants)
than those with placebo (189 participants) (SMD -0.42, 95% CI -0.69 to -0.14; partici-
pants = 380; studies = 4; $I^2$ = 38%).

*Nasal itchiness*. Pooled estimates from four studies [29, 44, 48, 51] showed no significant
differences in nasal itchiness scores between the medicinal plant-treated (191 participants) and
placebo arms (189 participants) (SMD -0.14, 95% CI -0.34 to 0.06; participants = 380; stud-
ies = 4; $I^2$ = 0%).

*Sneezing*. Pooled estimates from four studies [29, 44, 48, 51] showed no significant differ-
ences in sneezing scores between the medicinal plant-treated (191 participants) and placebo
arms (189 participants) (SMD -0.24, 95% CI -0.61 to 0.13; participants = 380; studies = 4; $I^2$ =
65%). There is substantial heterogeneity, as shown by the $I^2$ of 65%. One study was conducted
among children [51] using an intranasal formulation, while the remaining three [Insert Jung
2021] [29, 44, 48] investigated oral formulations of different medicinal plants in adults. The
duration of study varied from 14 to 42 days. We explored plausible factors that might account
for the heterogeneity observed in terms of population, intervention, comparison, and out-
comes, and found that by excluding one study [51] on children, the $I^2$ reduced to 0% with
pooled estimates favouring neither treatment groups (SMD -0.05, 95% CI -0.34 to 0.24; partici-
pants = 189; studies = 3; $I^2$ = 0%). Co-incidentally, this also reflects subgroup analysis by route
of administration (i.e., children with intranasal administration vs adults with oral administra-
tion). The remaining single study suggested that the medicinal plant may slightly improve
nasal congestion among children (SMD -0.63, 95% CI -0.93 to -0.34; participants = 191; stud-
ies = 1) (S2 Fig in S9 Appendix).

We found that age and/or route of administration to be plausible reasons that contributed
towards heterogeneity for rhinorrhoea and sneezing.

*Total eye symptoms*. One study [29] suggested at no improvements in total eye symptom
scores with the medicinal plant (MD -0.10, 95% CI -0.92 to 0.72; participants = 77;
studies = 1).

*Eye itchiness*. One study suggested at no improvements in eye itchiness scores with the
medicinal plant (MD 0.11, 95% CI -0.42 to 0.64; participants = 40; studies = 1).

*(b) Total nasal symptom score (changes in mean)*. Results from one study [33] suggested no difference in total nasal symptom scores (changes in mean) (MD 0.00, 95% CI -1.11 to 1.11; participants = 32; studies = 1) among those treated with the medicinal plant and antihistamine (S9 Fig in S7 Appendix).

*(c) Total reflexive and instantaneous nasal, eye, and throat symptom score*. Results from one study suggested no differences in total reflexive [39] and instantaneous nasal, throat and eye symptom scores (MD -0.35, 95% CI -1.36 to 0.66; MD 0.00, 95% CI -2.92 to 2.92; participants = 223; studies = 1) (S10 Fig in S7 Appendix).

*(d) Global symptoms assessment*. Results from one study [29] suggested no differences in global assessment scores (MD -0.02, 95% CI -0.69 to 0.65; participants = 77; studies = 1) but another [39] reported that participants treated with medicinal plant were more likely to report improvement in global symptoms (RR 7.01, 95% CI 2.86 to 17.16; participants = 220; studies = 1) (S11 and S12 Figs in S7 Appendix).

*(e) Rhinitis control assessment test (RCAT)*. Results from one study [29] suggested that the medicinal plant does not improve RCAT scores (MD -1.14, 95% CI -3.23 to 0.95; participants = 77; studies = 1) (S13 Fig in S7 Appendix).

*(f) Other symptoms*. Results from one study [29] suggested at no differences or improvements in ear symptoms, throat symptoms, headache, mental function, post nasal drip, and cough scores (Throat symptoms: MD 0.01, 95% CI -0.74 to 0.76; Ear symptoms: MD 0.00, 95% CI -0.76 to 0.76; Post nasal drip: MD -0.07, 95% CI -0.80 to 0.66; Headache: MD 0.26, 95% CI -0.42 to 0.94; Mental function: MD 0.17, 95% CI -0.59 to 0.93; Cough: MD 0.01, 95% CI -0.56 to 0.58; participants = 77; studies = 1) (S14-S19 Figs S7 Appendix).

*(g) Clinical signs*. One study [44] showed that the medicinal plant likely resulted in little to no differences in measurements of the participants' nasal cavities (Minimum cross sectional area of right nose: MD 0.05, 95% CI -0.02 to 0.12; Minimum cross sectional aera of left nose: MD 0.04, 95% CI -0.01 to 0.09; Volume estimates of right nasal cavity MD 0.75, 95% CI 0.14 to 1.36; Volume estimates of left nasal cavity MD 0.58, 95% CI 0.08 to 1.08; Distance from the nostril of the right nose: MD 0.03, 95% CI -0.10 to 0.16; Distance from the nostril of the right nose: MD -0.19, 95% CI -0.34 to -0.04; participants = 72; studies = 1) (S20 Fig in S7 Appendix).

Results from one study [51] which assessed the physical appearance of participants' nasal cavities suggested slight improvements in nasal mucosa oedema but not in inferior turbinate swelling with the medicinal plant while the evidence for its effect on nasal mucosal swelling is very uncertain (Inferior turbinate swelling score: MD -0.43, 95% CI -0.87 to 0.00; Nasal mucosa oedema score: MD -0.59, 95% CI -0.72 to -0.46; Nasal mucosa swelling score: MD -0.15, 95% CI -0.26 to -0.04; participants = 191; studies = 1) (S21 Fig in S7 Appendix).

Results from one study [33] suggested at improvements in PNIF among those treated with the medicinal plant (MD -12.00, 95% CI -14.08 to -9.92; participants = 32; studies = 1) while another [48] shows no difference in morning and afternoon PNIF. These two studies could not be pooled as different units and methods of measurement were used (S22 and S23 Figs in S7 Appendix).

*(h) QOL*. Results from one study [44] reported no differences in the total RQLQ scores and their individual components among those treated with the medicinal plant and antihistamine (Total: MD -0.10, 95% CI -0.56 to 0.36; Activity limitations: MD -0.40, 95% CI -1.01 to 0.21; Sleep problems: MD -0.05, 95% CI -0.62 to 0.52; Nose symptoms: MD -0.29, 95%

CI -0.91 to 0.33; Eye symptoms: MD 0.05, 95% CI -0.51 to 0.61; Non-nose/eye symptoms: MD -0.07, 95% CI -0.61 to 0.47; Practical problem: MD -0.02, 95% CI -0.61 to 0.57; Emotional function: MD -0.02, 95% CI -0.48 to 0.44; participants = 72; studies = 1) (S24 Fig in S7 Appendix).

Another single study which reported [51] on other QOL scores (unspecified) also similarly suggested no differences among medicinal plant and antihistamine treated participants (MD 0.20, 95% CI -0.44 to 0.84; participants = 77; studies = 1) (S25 Fig in S7 Appendix).

One study [37] reported SF-36 median scores of physical function, emotional function, vitality, mental health, general health, social functioning, and pain to be similar among those treated with standardised *P. hybridus* tablets and cetirizine for two weeks. Participants treated with *P. hybridus* tablets reported better scoring on physical function (Median = 100, Min-Max = 0–100) compared to cetirizine (Median = 75, Min-Max = 0–100). Both groups reported improvements in median scores from baseline in all domains. This study was not pooled as the team was not able to obtain sufficient data to reasonably estimate the mean and SD.

**Medicinal plant vs intranasal corticosteroid.**   Results from a small study [47] suggested that those treated with the medicinal plant may have slight worsening of total nasal symptom scores compared to those treated with intranasal corticosteroid (MD 0.29, 95% CI 0.03 to 0.54; participants = 21; studies = 1) (S26 Fig in S7 Appendix).

**Medicinal plant as add-on to standard treatment vs standard treatment.**   Six studies [23, 24, 26, 36, 47, 50] compared medicinal plants and conventional treatment combined against conventional treatment only.

*(a) Nasal and eye symptom score.* One study [24] reported no significant differences in total nasal symptoms, eye symptoms, and individual nasal symptoms scores when medicinal plant was added onto standard treatment (Total nasal symptom: MD -1.00, 95% CI -3.46 to 1.46; Rhinorrhoea: MD 0.00, 95% CI -0.74 to 0.74; Nasal congestion: MD -0.50, 95% CI -1.55 to 0.55; Nasal itchiness: MD -0.50, 95% CI -1.24 to 0.24;Sneezing: MD 0.13, 95% CI -0.61 to 0.87; Total eye symptom: MD -0.75, 95% CI -2.00 to 0.50; Watery eyes: MD -0.12, 95% CI -0.76 to 0.52; Itchy eyes: MD -0.50, 95% CI -1.14 to 0.14; Red eyes: (MD -0.13, 95% CI -0.37 to 0.11; participants = 16; studies = 1) (S27 Fig in S7 Appendix).

On the contrary, another single study [47] suggested at a slight worsening of total nasal symptom scores with medicinal plants as add-ons, reported as changes in mean (MD 0.31, 95% CI 0.05 to 0.57; participants = 21; studies = 1) (S28 Fig in S7 Appendix). The certainty of evidence is low for both contradicting findings.

*(b) Sino-nasal outcome test (SNOT-22).* Pooled estimates from four studies [23, 26, 36, 50] showed that those treated with medicinal plants combined with standard treatment may have significantly better SNOT-22 scores (98 participants) compared to conventional treatment alone (89 participants) (MD -8.61, 95% CI -14.74 to -2.74; participants = 187; studies = 4; $I^2$ = 64%) but the evidence is very uncertain. These studies were considered to have high heterogeneity (Fig 12). There is substantial heterogeneity, as shown by the $I^2$ of 64%. All four studies investigated different medicinal plants, two [36, 50] administered the herbal intervention intranasally. The standard treatment (comparator) was also different for all three studies. The study duration varied from 28 to 60 days. Route of administration was found to be a plausible source of heterogeneity.

Subgroup-analyses by route of administration showed that medicinal plants given intranasally significantly improved SNOT-22 when added to standard treatment (MD -7.47, 95% CI

-10.75 to -4.18; participants = 124; studies = 2; I² = 21%). This was not seen when medicinal plants were given orally (Fig 12).

*(c) Clinical signs.* Results from one study [36] suggested at improvements in Lund-McKay scores and modified Lund Kennedy scores with an add-on medicinal plant (Lund-McKay score MD -1.88, 95% CI -2.44 to -1.32; Modified Lund Kennedy score: MD -1.89, 95% CI -2.40 to -1.38; participants = 65; studies = 1) while another single study [24] reported no improvements in nasal airway resistance (MD -0.05, 95% CI -0.29 to 0.19; participants = 16; studies = 1) (S29-S31 Figs in S7 Appendix). Sample size is small in both studies.

## Plausible factors of heterogeneity

For most of the outcomes with sufficient studies for subgroup analyses, we could not identify plausible factors of heterogeneity based on population, intervention, comparator, and outcome.

When comparing medicinal plants against placebo, the use of additional rescue medications such as antihistamines appears to be a plausible factor for heterogeneity in likelihood of reporting improvements in global symptoms, but not for special nasal scores on sneezing.

When comparing medicinal plants against antihistamine, age and/or route of administration appeared to be plausible reasons that contributed towards heterogeneity for rhinorrhoea, nasal congestion, and sneezing.

## Certainty of evidence (GRADE)

The certainty for most outcomes evaluated in this review was very low to low. The main downgrading factors were unclear risk of bias and its potential effect on the outcomes which were mostly subjective based on perception on individual symptoms, imprecision due to wide CI and small sample size as well as inconsistency presented as high I² values. The details of GRADE assessment are presented in S1 to S4 Tables in S10 Appendix.

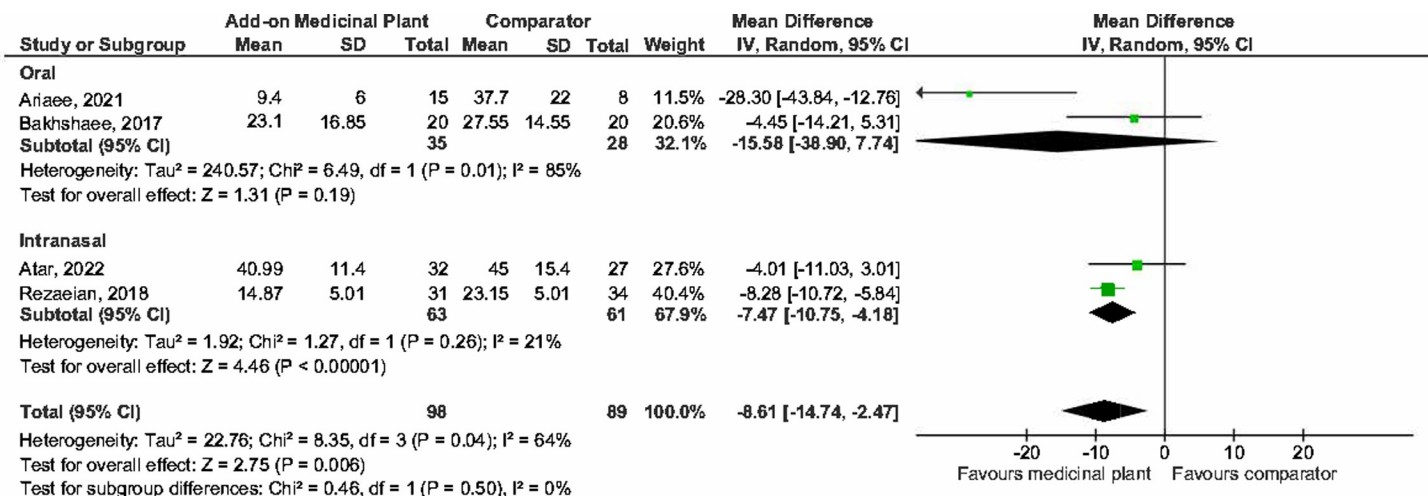

**Fig 12. Forest plot comparison of medicinal plant (add-on to conventional therapy) vs conventional therapy alone for SNOT-22.**

### Reporting and publication bias

As there were insufficient studies of the same outcome and meta-analysis to generate a meaningful funnel plot, this was not performed. Among the 28 included studies, only 12 specified their clinical trial registry details. As for unpublished trial data, there were a total of 3 trial registers of completed studies without publicly available data.

### Reporting quality

The reporting quality of included studies is presented in S11 Appendix, based on the criteria set out in the CONSORT extension for reporting herbal medicines, item No.4, which focuses on the details needed when reporting on a medicinal plant preparation as an investigational product. Overall, it was observed that there is insufficient reporting on the details recommended for medicinal plant preparations. Only 11 studies (37.9%) stated the full Latin binomial name together with botanical authority and family name of the medicinal plant when describing the investigational product or formulation. In fact, three studies [27, 32, 47] did not mention the scientific name of the medicinal plant altogether. Only a single study mentioned that the product used was authorised (licensed, registered) in the country in which the study was conducted [33]. Most included studies provided details on part of plants used as well as its type e.g., extract, powder, etc. However, details on the formulation including type and concentration of the extract, and quantity of each ingredient in the formulation were not sufficiently described in most studies. Few studies provided sufficient details of method of authentication, special/purity testing, chemical fingerprinting and methods used, as well as justification for dosage (1 to 4 studies in each domain). No studies reported the involvement of an alternative therapy practitioner.

### Safety

Overall, mild to moderate adverse events were reported in the included studies. Commonly reported adverse events in the treatment group include headache, dizziness, and gastrointestinal disturbances, which were also similarly reported in the control groups of most studies, with the exception of one study which reported high numbers of eructation in the medicinal plant arm (n = 26 vs n = 6). This study investigated the effects of *Zingiber officinale* Roscoe [44]. Raised liver enzymes, which resolved within two weeks (one case with raised Alanine transaminase (ALT)) or did not lead to discontinuation of treatment (one case with raised ALT and Aspartate transaminase (AST)), were reported in two patients who were given fermented red ginseng [32] and *Pinus pinaster* Ait [43]. respectively. A different study on Korean red ginseng reported no laboratory abnormalities though the authors did not specify which laboratory parameters were monitored for safety [48]. The details of adverse events for each included study are presented in Table 2.

## Discussion

This review found no clear evidence that medicinal plants can substantially improve most of the commonly reported clinical outcome measures of allergic rhinitis, including scores for overall and individual nasal and eye symptoms when compared against placebo or antihistamine. Pooled analysis suggest that medicinal plants may improve total RQLQ scores when compared against placebo while single studies suggested no improvements in RQLQ and other generic QOL scores when comparing medicinal plants against antihistamine. As add-ons to conventional treatment, medicinal plants appeared to improve SNOT-22 scores when given intranasally, but the number of studies is small. Specifically, when compared against placebo,

**Table 2. Safety of medicinal plants for allergic rhinitis.**

| Author | AE in medicinal plant | AE in comparator | Liver function | Renal function |
|---|---|---|---|---|
| Achararit, 2023 | No adverse events reported. | No adverse events reported. | No abnormalities | No abnormalities |
| Atar, 2022 | NR | NR | NR | NR |
| Ariaee, 2021 | NR | NR | NR | NR |
| Jung, 2021 | Abdominal pain (n = 2), nausea (n = 1) | No adverse effects reported | NR | NR |
| Yusin, 2021 | Gastrointestinal disturbances (n = 1) leading to withdrawal of subjects | Gastrointestinal disturbances (n = 3) leading to withdrawal of subjects | NR | NR |
| Yamprasert, 2020 | Eructation (n = 26), drowsiness (n = 1), dry mouth (n = 4), dry throat (n = 4), fatigue (n = 1), dizziness (n = 1) | Eructation (n = 6), drowsiness (n = 9), dry mouth (n = 5), dry throat (n = 7), keen nose (n = 2), fatigue (n = 4), dizziness (n = 3), constipation (n = 3) | No abnormalities | No abnormalities |
| Derakhshan, 2019 | Nausea (n = 2), abdominal discomfort (n = 3) | Fexofenadine: Drowsiness (n = 4) | NR | NR |
| Steels, 2019 | Worsening of allergic rhinitis symptoms (n = 2), significant decrease in white blood cell and neutrophil counts compared to placebo (within normal laboratory range) | Nostril bleeding, itchy skin | No abnormalities | No abnormalities |
| Arpornchayanon, 2019 | Dizziness, fatigue, headache, somnolence, rashes, decreased strength of hair root, nausea, and dyspepsia. No difference between groups | Dizziness, fatigue, headache, somnolence, rashes, decreased strength of hair root, nausea, and dyspepsia. No difference between groups | NR | NR |
| Rezaeian, 2018 | No adverse effects reported | No adverse effects reported | NR | NR |
| Hajiheydari, 2017 | No side effects reported | No side effects reported | NR | NR |
| Bakhshaee, 2017 | No serious, deleterious adverse effects reported. | No serious, deleterious adverse effects reported. | NR | NR |
| Walanj, 2014 | Cough, fever, headache, body aches, throat irritation. (n = 9) | Cold, earache, fatigue, itching in eyes, throat irritation, watery eyes. (n = 14) | No abnormalities | No abnormalities |
| Yonekura, 2011 | No adverse events reported | No adverse events reported | NR | NR |
| Jung, 2011 | Mild rise in ALT (n = 1) Raised liver enzymes resolved in 2 weeks. No significant difference between groups | Mild rise in ALT (n = 1), mild rise in ALT and bilirubin (n = 1) Raised liver enzymes resolved in 2 weeks. No significant difference between groups | Mild rise in ALT | NR |
| Matkovic, 2010 | Rhinosinusitis (n = 4), pharyngitis (n = 7), enterocolitis (n = 1), nausea (n = 1), lacunar angina (n = 1), vulvitis (n = 1). Reported as unrelated to treatment. Adverse event in each treatment/placebo group not specified | Rhinosinusitis (n = 4), pharyngitis (n = 7), enterocolitis (n = 1), nausea (n = 1), lacunar angina (n = 1), vulvitis (n = 1). Reported as unrelated to treatment. Adverse event in each treatment/placebo group not specified | NR | NR |
| Wilson, 2010 | Headache, dizziness, common cold, dry mouth (n = 15); raised ALT and AST (n = 1), did not require discontinuation of treatment | Headache, dizziness, common cold, dry mouth (n = 17); severe vertigo (n = 1), led to withdrawal of subject | Raised ALT and AST | NR |
| Wu, 2009 | Localised nose pain (details of each group not specified) | Localised nose pain (details of each group not specified) | NR | NR |
| Cingi, 2008 | NR | NR | NR | NR |
| Yoshimura, 2007 | Cold and diarrhoea (details of each group not specified) | Cold and diarrhoea (details of each group not specified) | NR | NR |
| Schapowal, 2005 | Any adverse event (n = 8), sedation complex (n = 6), common cold (n = 1), fever (n = 1), nausea (n = 1), sinus pain (n = 1) | Fexofenadine: Any adverse event (n = 10), sedation complex (n = 4), common cold (n = 3), nausea (n = 1), sinus pain (n = 1), diarrhoea (n = 1), headache (n = 1), limb pain (n = 1), stomach pain (n = 1) Placebo: Any adverse event (n = 7), sedation complex (n = 3), common cold (n = 2), uterine abrasion (n = 1), headache (n = 1), light sensitivity (n = 1) | NR | NR |
| Badar, 2005 | Nasal pain (n = 2), headache (n = 1) | None | NR | NR |
| Schapowal, 2004 | HD: Gastrointestinal upset (n = 3), nausea (n = 2), skin rash (n = 1), hot flushes (n = 1), diarrhoea (n = 1), influenza (n = 1) LD: Gastrointestinal upset (n = 1) | Gastrointestinal upset (n = 1), nausea (n = 1), skin rash (n = 1) | NR | NR |

*(Continued)*

**Table 2.** (Continued)

| Author | AE in medicinal plant | AE in comparator | Liver function | Renal function |
|---|---|---|---|---|
| Gray, 2004 | Headache (n = 2), nausea (n = 1), painful eyes (n = 1), indigestion (n = 1), wheeze (n = 2), itch (n = 1) | Headache (n = 3), painful eyes (n = 1), painful ears (n = 1), indigestion (n = 1), rash (n = 1) | NR | NR |
| Takano, 2004 | No adverse events reported. | No adverse events reported. | No abnormalities | No abnormalities |
| Lee, 2004 | NR | NR | NR | NR |
| Schapowal, 2002 | Any adverse event (n = 10). Details not specified | Cetrizine: Any adverse event (n = 11); two thirds were drowsiness and fatigue | NR | NR |
| Bernstein, 2002 | Headache, abdominal pain, and sore throat. No significant difference between groups | Headache, abdominal pain, and sore throat. No significant difference between groups | NR | NR |
| Mittman, 1990 | Intensification of allergic symptoms (n = 2), gastrointestinal discomfort (n = 5) | Gastrointestinal discomfort (n = 5) | NR | NR |

Adverse event (AE); High dose (HD); Low dose (LD); Alanine transaminase (ALT); Aspartate transaminase (AST); NR (Not reported)

participants who were treated with medicinal plants were more likely to report improvements in individual symptoms of rhinorrhoea and nasal itchiness, measured as responder rates. A single study with a small sample size compared the effects of a medicinal plant against intranasal corticosteroids and found that the medicinal plant treated group had slightly worse total nasal symptom scores. Other less commonly measured outcomes such as clinical signs, other non-eye/nose symptoms, rescue medication use, impairment in activities, and satisfaction scores were assessed in single studies. The certainty of evidence for most pooled outcomes evaluated in this review was very low to low. The main downgrading factors were unclear risk of bias, imprecision due to wide CI and small sample size, as well as inconsistency presented as high $I^2$ values with no major identifiable plausible causes of heterogeneity.

The findings of this review are different from previously published systematic reviews which reported significant improvements in total nasal and eye symptoms as well as QOL scores among participants who were treated with herbal medicine and Chinese herbal medicines [11, 15, 52]. The objective and inclusion criteria of these previously published reviews which encompassed single herbs or herb mixtures and comparators were distinctively different from our review as we focused on studies of single medicinal plants analysed according to categories of comparators with detailed breakdowns of different outcome measures. The certainty of evidence ratings, which was performed in the present review but not in previously published reviews, might also have influenced our conclusions. Substantial heterogeneity during meta-analyses was found as in previously published reviews [11, 15]. In our review, age and route of administration were additionally identified as plausible factors that contribute towards heterogeneity. Due to the small number of studies investigating the same plant, we were only able to assess plant type as a plausible factor of heterogeneity in one sub-group analysis and found that it did not have a substantial effect. This finding therefore requires confirmation from further studies.

As for safety, similar to previous reports [15], herbal interventions appeared to be reasonably well tolerated among participants with mild to moderate but resolvable adverse events reported, even in the two cases of liver enzyme derangements. One plant in particular, *Z. officinale*, was associated with four times higher cases of eructation vs loratadine [44]. However, insufficient reporting on the quality and constituents of the herbal formulations precluded a more detailed evaluation on their safety profile.

Overall, there was a low adherence to the reporting guidelines for herbal trials. Insufficient details especially on the investigational product in terms of authentication, quality, and quantitative analysis hinders in-depth assessment of the true contribution of an herbal intervention, especially in construing mechanisms of action or a dose-response relationship. This piece of missing information is also important in aiding the translation and extrapolation of the findings to other similar preparations of the same reported plant. This review concurs with the "no recommendations" recommendation by the American Academy of Otolaryngology-Head and Neck Surgery Foundation due to insufficient information available on the quality and quantity analysis of herbal interventions in randomised clinical trials, despite the guideline being published six years ago [9].

A noteworthy observation from this review is the variety of outcome measures and scales used especially for overall global assessment, total, and specific nasal symptoms, some with conflicting findings. Despite being scored based on validated or standard questionnaires, allergic rhinitis symptoms and quality of life assessment is subjective as judgements can be affected by an individual's expectations and perceptions. Individuals may differ in their views of how certain symptoms are more important than others, depending on their impact on important activities of daily living or work for the individual in question. Therefore, although medicinal plants may not clearly improve overall symptoms, patients' preference for choice of treatment for specific symptoms should be given sufficient consideration. This is in-line with the current management approach that takes into account patients' preference [5]. Continued development of herbal formulations with sufficiently reported quality and safety data is essential for the future evaluation and application of herbal medicine for allergic rhinitis. In addition, knowledge and understanding of the intricate properties of herbal preparations among recommending physicians is also key to ensure safe administration of herbal products.

## Limitations

Although this review did not exclude non-English articles, we did not explicitly search in Chinese language journal databases due to resource restrictions. Chinese herbs are especially popular, evidenced by the many systematic reviews on this traditional practice [52–54]. However, we included one [51] Chinese language paper. Given that Chinese herbal medicine is often given in the form of mixtures, it is less likely that this review paper missed out on significant Chinese single-herb papers. A systematic review protocol specifically on Chinese medicine for allergic rhinitis has been published, indicative of an ongoing review [55], which would provide evidence specific to Chinese herbal medicine.

## Conclusion

There is no clear evidence that medicinal plants improve overall clinical symptoms when compared to placebo or antihistamine while limited evidence suggest improvements in RQLQ specifically when compared against placebo. Limited evidence also suggest improvement in related measures when medicinal plants are used as add-ons to conventional treatments. There is a need for improved reporting of herbal trials to allow for critical assessment of the effects of each individual herbal formulation in well-designed future clinical studies.

## Supporting information

**S1 Appendix. Full search strategy.**
(DOCX)

**S2 Appendix. Data extraction table.**
(DOCX)

**S3 Appendix. Details on data processing during analyses.**
(DOCX)

**S4 Appendix. List of excluded studies after full text screening.**
(PDF)

**S5 Appendix. Characteristics of included study.**
(DOCX)

**S6 Appendix. Unpublished clinical trials.**
(DOCX)

**S7 Appendix. Supplementary figures of forest plots.**
(DOCX)

**S8 Appendix. Forest plot comparison of medicinal plant vs placebo for RQLQ (without exclusion of Steels, 2019).**
(DOCX)

**S9 Appendix. Subgroup analysis (Medicinal plant vs oral antihistamines- symptom scores).**
(DOCX)

**S10 Appendix. Summary of findings table.**
(DOCX)

**S11 Appendix. Reporting quality of herbal preparations in randomised controlled trials.**
(PDF)

**S12 Appendix. PRISMA 2020 checklist for systematic review.**
(DOCX)

## Acknowledgments

We would like to thank the Director General of Health Malaysia, Deputy Director General of Health Malaysia (Research & Technical Support), and the Director of Institute for Medical Research for their support and permission to publish this article.

## Author Contributions

**Conceptualization:** Xin Yi Lim, Terence Yew Chin Tan.

**Data curation:** Xin Yi Lim, Mei Siu Lau, Nor Azlina Zolkifli, Umi Rubiah Sastu@Zakaria, Nur Salsabeela Mohd Rahim, Terence Yew Chin Tan.

**Formal analysis:** Xin Yi Lim, Mei Siu Lau, Nai Ming Lai.

**Investigation:** Xin Yi Lim, Mei Siu Lau, Nor Azlina Zolkifli, Umi Rubiah Sastu@Zakaria, Nur Salsabeela Mohd Rahim, Terence Yew Chin Tan.

**Methodology:** Xin Yi Lim, Nai Ming Lai, Terence Yew Chin Tan.

**Project administration:** Xin Yi Lim.

**Software:** Xin Yi Lim.

**Supervision:** Nai Ming Lai.

**Validation:** Nai Ming Lai.

**Writing – original draft:** Xin Yi Lim.

**Writing – review & editing:** Xin Yi Lim, Mei Siu Lau, Nor Azlina Zolkifli, Umi Rubiah Sastu@Zakaria, Nur Salsabeela Mohd Rahim, Nai Ming Lai, Terence Yew Chin Tan.

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
