## [Decision Letter · Decision Letter 0]

12 Dec 2023

PONE-D-23-37729Medicinal plants for allergic rhinitis: a systematic review and meta-analysisPLOS ONE

Dear Dr. Lim,

Thank you for submitting your manuscript to PLOS ONE. After careful consideration, we feel that it has merit but does not fully meet PLOS ONE’s publication criteria as it currently stands. Therefore, we invite you to submit a revised version of the manuscript that addresses the points raised during the review process.

 Please submit your revised manuscript by Jan 26 2024 11:59PM. If you will need more time than this to complete your revisions, please reply to this message or contact the journal office at plosone@plos.org. Please include the following items when submitting your revised manuscript:A rebuttal letter that responds to each point raised by the academic editor and reviewer(s). You should upload this letter as a separate file labeled 'Response to Reviewers'.A marked-up copy of your manuscript that highlights changes made to the original version. You should upload this as a separate file labeled 'Revised Manuscript with Track Changes'.An unmarked version of your revised paper without tracked changes. You should upload this as a separate file labeled 'Manuscript'.

We look forward to receiving your revised manuscript.

Kind regards,

Rajeev Singh

Academic Editor

PLOS ONE

Journal Requirements:

2. We note that this manuscript is a systematic review or meta-analysis; our author guidelines therefore require that you use PRISMA guidance to help improve reporting quality of this type of study. Please upload copies of the completed PRISMA checklist as Supporting Information with a file name “PRISMA checklist”.

Reviewers' comments:

Reviewer's Responses to Questions

**Comments to the Author**

1. Is the manuscript technically sound, and do the data support the conclusions?

Reviewer #1: Yes

Reviewer #2: Partly

Reviewer #3: Yes

2. Has the statistical analysis been performed appropriately and rigorously? 

Reviewer #1: Yes

Reviewer #2: Yes

Reviewer #3: Yes

3. Have the authors made all data underlying the findings in their manuscript fully available?

Reviewer #1: Yes

Reviewer #2: Yes

Reviewer #3: Yes

4. Is the manuscript presented in an intelligible fashion and written in standard English?

Reviewer #1: Yes

Reviewer #2: Yes

Reviewer #3: Yes

5. Review Comments to the Author

Reviewer #1: This is an excellent review on herbal medicine for allergic rhinitis.

I would like to suggest the following:

1. The introduction is too lengthy. Usually 1 and 1/2 pages are usually needed. There is no need to explain the type of allergic rhinitis and the issue of under diagnosis unless they are related to under study subject.

2. The review and meta-analysis can be better understood and beneficial if the medicinal plants can be grouped into topical and systemic route of administration.

Reviewer #2: This systematic review and meta-analysis focuses on the effectiveness and safety of individual medicinal plants in allergic rhinitis,trying to emphasize medicinal plants as a significant complementary and alternative treatment option based on its reasonable clinical effectiveness, good patient compliance and minimal adverse reactions.However,the manuscript lacks originality and offers no novel insights.Therefore a rejection of the manuscript is suggested for the following reasons:

1.Part of the data needs to be verified.The conclusion states, "From the initial 1,523 articles identified, 39 articles were included, comprising 29 published randomized controlled trials. We identified 10 unpublished or ongoing registered trials." However, Figure 1 and the abstract mentioned only 29 cases.

2.Although there's no length restriction for manuscripts writing, concise presentation style and discussion based on the findings are encouraged.Parts of the manuscript are repetitive and redundant, particularly in the sections on Search Strategy, Study Selection, and Data Collection.

3.The criteria for inclusion and exclusion of literature are crucial for the outcomes of the meta-analysis and the overall quality of the article. I recommend adding these criteria in the main text rather than referring to Appendix S1.

4.Due to different language barrier in relation to literature availability in journals published in Chinese,this study has not thoroughly covered Chinese literature. Therefore it is suggested to supplement these data.

5.The study involved a variety of medicinal plants and their components. This diversity could lead to a significant bias in the research outcomes, making it challenging to draw reliable and unified conclusions. The variation in medicinal plants ingredients across the studies should be addressed to ensure the validity and applicability of the findings

6.Various measures and scales were used in the study, particularly for overall assessment，general and specific nasal symptoms, some contradictory results (as seen in Figures 4, 5, and 8) were reported. Although explanations have been provided for these discrepancies, it seems that the results were not so reliable.

Reviewer #3: This Systematic of medicinal plant has been conducted with good quality. The search term and database searching are appropriate. The methodology of data extraction is clear with good standard operating protocol (SOP). The gap of knowledge from previous study has been clearly stated. Data and result of the study are clearly shown by graphs and tables. The significant finding from this study is single medicinal plants may improve overall nasal symptoms comparing to placebo or antihistamine with very low to low certainly. But it can improve quality of life when comparing against placebo. This review study also mention about some limitations such as possibility of missing some Chinese single-herb paper. Overall, this systematic review is conducted with the excellent quality, can be utilized in the clinical practice guideline and implementation.

6. PLOS authors have the option to publish the peer review history of their article (what does this mean?). If published, this will include your full peer review and any attached files.

Reviewer #1: No

Reviewer #2: No

Reviewer #3: **Yes: **Pongsakorn Tantilipikorn MD PhD

---

## [Author Response · Author response to Decision Letter 0]

4 Jan 2024

Dear Editor,

Thank you for considering the publication of our manuscript. 

Please find this letter of response to reviewers in three parts. I will first start with Reviewer 1 and 3, followed by reviewer 2. All responses are in blue.

Reviewer 1

This is an excellent review on herbal medicine for allergic rhinitis.

I would like to suggest the following:

1. The introduction is too lengthy. Usually 1 and 1/2 pages are usually needed. There is no need to explain the type of allergic rhinitis and the issue of under diagnosis unless they are related to under study subject.

Response: Thank you for the suggestions. I have edited as recommended.

2. The review and meta-analysis can be better understood and beneficial if the medicinal plants can be grouped into topical and systemic route of administration.

Response: Thank you for the suggestions. We agree that there is great value to presenting results as such. However, upon further looking into the data, we realised that out of the 29 papers included for analysis, only 5 were intranasal. Upon further breakdown, this results in maximum of 2 articles with intranasally administered medicinal plants intervention in selected individual comparisons (and some none at all). Due to the large amount of separate analysis for different outcomes and subgroup analyses, we would like to request your understanding that we are unable to perform additional analyses on this small group of results. In addition, route of administration is one of the key plausible factors for heterogeneity when pooling studies, that we have tried to explored without any significant findings. Therefore, when considering the additional information, we do not think that it will change the results. We truly appreciate your positive feedback and we will keep in mind of this option for our future manuscripts. We really want to thank you. 

Reviewer 3

This Systematic of medicinal plant has been conducted with good quality. The search term and database searching are appropriate. The methodology of data extraction is clear with good standard operating protocol (SOP). The gap of knowledge from previous study has been clearly stated. Data and result of the study are clearly shown by graphs and tables. The significant finding from this study is single medicinal plants may improve overall nasal symptoms comparing to placebo or antihistamine with very low to low certainly. But it can improve quality of life when comparing against placebo. This review study also mention about some limitations such as possibility of missing some Chinese single-herb paper. Overall, this systematic review is conducted with the excellent quality, can be utilized in the clinical practice guideline and implementation.

Response: Thank you very much for the positive response. We are very pleased that the message that we wanted to express through this review was comprehended clearly by readers. 

Reviewer 2: Rebuttal: Recommendation for rejection by reviewer 2

I am writing in response to the recommendation for rejection by Reviewer 2, as opposed to the supportive recommendations from reviewer 1 and 3 for publication. Please find my response to each of the comments as such:

This systematic review and meta-analysis focuses on the effectiveness and safety of individual medicinal plants in allergic rhinitis,trying to emphasize medicinal plants as a significant complementary and alternative treatment option based on its reasonable clinical effectiveness, good patient compliance and minimal adverse reactions. However,the manuscript lacks originality and offers no novel insights. Therefore a rejection of the manuscript is suggested for the following reasons:

Response: The research gap that underscores the undertaking of this systematic review and meta-analysis are explained in the introduction, which is well-understood by other reviewers. However, I will try to reiterate and emphasize the important points to support the importance and impact of our manuscript

1. Research impact: Herbal medicine is popular and has potential to be the preferred choice of management for allergic rhinitis, by patients. As explained in the introduction, patients’ own empowerment and choice of treatment plays an important role in the management of allergic rhinitis. This is also in-line with clinical practice guideline’s recommendation. Studies have shown poor compliance among patients using conventional medicine. On the other hand, data supports the popularity of complementary and alternative medicine, including herbal medicine among these patients. Due to that, numerous clinical trials have been conducted on herbal medicine (as shown in our findings), as single medicinal plant, mixture, in combination with other complementary and alternative therapy. 

2. Originality: We agree that several reviews have been conducted in the past. However, we cannot agree on the comment that our study does not provide new insights to the available evidence. Here are the unique strengths of our paper:

a. Single medicinal plant as intervention: In the past, no systematic review and/or meta-analysis were focused on single plants. Therefore, this review attempted to reduce the heterogeneity in intervention to the best of our ability by limiting the studies to only single medicinal plants, which is often an important factor when pooling studies across the vast range of complementary and alternative medicine. In addition, we further divided the outcome analysis according to comparator. To completely eliminate this heterogeneity (due to interventions), our original intention was to pool studies by plant type as laid down in the registered protocol on PROSPERO. However, due to insufficient studies to do so, we had to deviate from the protocol and this has been explained clearly in Appendix S3 as originally submitted. 

b. New studies: Due to the popularity of complementary and alternative medicine, the research landscape is quickly expanding, with new studies and results emerging quite significantly over the years. During the submission of this review, the most recent review on herbal medicine that could be found is a systematic review and meta-analysis by Hoang et al. (published in 2021) on herbal medicine for allergic rhinitis which collated evidence of herbal medicine use (as single herbs, mixture of herbs, or in combination with procedure-based therapies, including Traditional Chinese Medicine) from 32 randomised controlled trials. The latest search for this published review was Feb 2020 and included papers up until 2019. From our findings, we have an additional 6 papers (published since 2020) to our most recent research date of 18th July 2023. In such instances especially for evidence of low certainty, new studies have the potential to significantly change previous findings of pooled analyses. 

c. Certainty of evidence: The latest review (Hoang 2021) did not grade the certainty of evidence, which was another research gap that we identified in the overall literature (as part of our background work before undertaking this study) and have attempted to fill. Certainty of evidence is an important recommended step in evidence-based medicine synthesis by the Cochrane group. A well-known example is as seen in the COVID-19 literature synthesis era when pooling evidence (see paper Popp 2021 re: ivermectin). This paper, amongst many others, demonstrated the importance of grading the evidence in addition to pooling data. Grading of evidence certainty can completely change how recommendations are made, despite pooled data showing statistically significant differences. 

d. Outcomes assessed: Our paper covered 20 efficacy outcomes in addition to safety, which has never been collated to such extend of comprehensiveness before. This is also one of the main contributing factors to the length of this manuscript. As we are guided, and historically trained by a senior investigator with extensive evidence in conducting Cochrane systematic reviews, we intended to comprehensively review all available evidence to the best of our ability and uphold the highest standards of evidence synthesis.

1.Part of the data needs to be verified.The conclusion states, "From the initial 1,523 articles identified, 39 articles were included, comprising 29 published randomized controlled trials. We identified 10 unpublished or ongoing registered trials." However, Figure 1 and the abstract mentioned only 29 cases.

Response: Our apologies for the confusion. Throughout the results analysis, 29 studies are included in the qualitative analysis and 27 in the quantitative. This has always been consistent. The 10 unpublished trials were never included in the analysis (as there were no data to analyse). We have however presented some general information on the status of these trials in Appendix S6 instead. We have amended the wordings in results and Figure 1 to make this point clearer.

2.Although there's no length restriction for manuscripts writing, concise presentation style and discussion based on the findings are encouraged.Parts of the manuscript are repetitive and redundant, particularly in the sections on Search Strategy, Study Selection, and Data Collection.

Response: We have combined the section Search Strategy and Study Selection. Overall, we removed the names of the investigators to avoid lengthiness of the manuscript for these sections. Everything else were deem necessary to be included to us to ensure transparency of the steps that we have taken to undertake this extensive systematic review and meta-analysis. We welcome specific comments on which sentence or information is redundant to the reviewer for further improvement. 

3.The criteria for inclusion and exclusion of literature are crucial for the outcomes of the meta-analysis and the overall quality of the article. I recommend adding these criteria in the main text rather than referring to Appendix S1.

Response: Thank you for your suggestion. We have amended as recommended.

4.Due to different language barrier in relation to literature availability in journals published in Chinese,this study has not thoroughly covered Chinese literature. Therefore it is suggested to supplement these data.

Response: Thank you for your suggestion. We are well aware of the limitations on language for this review. This has been discussed in the limitations in the originally submitted manuscript. We do not have the resources to cover Chinese literature databases, as much as we would like to. However, we would like to highlight that most Chinese herbal medicines are multi-herbal or multi-ingredient in nature which therefore would not have been included according to our inclusion and exclusion criteria. This can be seen in the List of excluded studies after full text screening (Appendix S4), whereby 5 Chinese herbal medicine studies were excluded due to being mixture (and not due to language). In addition to that, in our experience, Chinese herbal medicine, when used within the principles of Traditional Chinese Medicine, is very much unique on its own. This is because of the substantial differences in the underlying principles of Western Medicine and Chinese Medicine philosophy, including approaches to management and symptoms assessments. We believe that Chinese Medicine warrants a separate review on its own due to its uniqueness. 

5.The study involved a variety of medicinal plants and their components. This diversity could lead to a significant bias in the research outcomes, making it challenging to draw reliable and unified conclusions. The variation in medicinal plants ingredients across the studies should be addressed to ensure the validity and applicability of the findings

Response: Thank you for your comment. As herbal researchers, we completely understand the concern on plant types, formulation, and quality. As explained earlier, and in Appendix S3, to completely eliminate this heterogeneity, our original intention was to pool studies by plant type as laid down in the registered protocol on PROSPERO. However, due to insufficient studies to do so, we had to deviate from the protocol. However, where ever feasible e.g. analysis Medicinal plants vs placebo-> (a) nasal and eye symptom score (post treatment means)-> nasal congestion, we attempted to pool the same plant for subgroup analysis but did not yield any significant results. We also have explored intervention type, wherever possible, as a plausible factor for heterogeneity but did not identify any. This approach of ours has been consistently mentioned throughout for each outcome and comparison analysed. Furthermore, we also assess the quality of reporting on quality-related details (including formulation related details) in Appendix S11 with the commentary in section: Reporting quality. It is quite clear that there is insufficient reporting on such details in the individual trials. Our conclusion also comments on this lack of reporting on botanical-related quality information.

6.Various measures and scales were used in the study, particularly for overall assessment，general and specific nasal symptoms, some contradictory results (as seen in Figures 4, 5, and 8) were reported. Although explanations have been provided for these discrepancies, it seems that the results were not so reliable.

Response: Thank you for noticing. Due to these discrepancies, and our attempts to identify plausible factors for heterogeneity, subgroup analyses, as well as in combination with our assessment on reporting quality, we have consistently (and originally) concluded that the evidence is of very low to low certainty. It can be a disappointment to some (or not) that after the huge tasks undertaken, results are conflicting. We have however, carefully and comprehensively conducted this analysis and it does indeed show that more work needs to be done to provide better evidence to recommend herbal medicine for allergic rhinitis, as we have stated in the conclusion. This is the message that we want to send across.

We standby the value of our manuscript, though lengthy, is comprehensive and very detailed. It has been undertaken according to recommended steps as per the Cochrane Evidence Synthesis recommendations and we believe it is worth publishing. We would really appreciate if you could consider our rebuttal points, and also take into account of the views of other reviewers while exercising your professional judgement on the extensive work that has been undertaken.

We sincerely thank you for the chance and opportunity given here.

Kind regards,

Xin Yi

---

## [Decision Letter · Decision Letter 1]

15 Jan 2024

Medicinal plants for allergic rhinitis: a systematic review and meta-analysis

PONE-D-23-37729R1

Dear Dr. Lim,

We’re pleased to inform you that your manuscript has been judged scientifically suitable for publication and will be formally accepted for publication once it meets all outstanding technical requirements.

Kind regards,

Rajeev Singh

Academic Editor

PLOS ONE

Additional Editor Comments (optional):

Reviewers' comments:

Reviewer's Responses to Questions

**Comments to the Author**

1. If the authors have adequately addressed your comments raised in a previous round of review and you feel that this manuscript is now acceptable for publication, you may indicate that here to bypass the “Comments to the Author” section, enter your conflict of interest statement in the “Confidential to Editor” section, and submit your "Accept" recommendation.

Reviewer #1: All comments have been addressed

2. Is the manuscript technically sound, and do the data support the conclusions?

Reviewer #1: Yes

3. Has the statistical analysis been performed appropriately and rigorously? 

Reviewer #1: Yes

4. Have the authors made all data underlying the findings in their manuscript fully available?

Reviewer #1: Yes

5. Is the manuscript presented in an intelligible fashion and written in standard English?

Reviewer #1: Yes

6. Review Comments to the Author

Reviewer #1: (No Response)

7. PLOS authors have the option to publish the peer review history of their article (what does this mean?). If published, this will include your full peer review and any attached files.

Reviewer #1: **Yes: **Prof Dr Baharudin Abdullah

---

## [Editor Report · Acceptance letter]

12 Feb 2024

PONE-D-23-37729R1 

PLOS ONE

Dear Dr. Lim, 

I'm pleased to inform you that your manuscript has been deemed suitable for publication in PLOS ONE. Congratulations! Your manuscript is now being handed over to our production team.

Kind regards, 

on behalf of

Dr. Rajeev Singh 

Academic Editor

PLOS ONE